# DICE: Diversity in Deep Ensembles via Conditional Redundancy Adversarial Estimation

**Alexandre Rame**
Sorbonne Université
Paris, France
`alexandre.rame@lip6.fr`

**Matthieu Cord**
Sorbonne Université & valeo.ai
Paris, France
`matthieu.cord@lip6.fr`

## Abstract

Deep ensembles perform better than a single network thanks to the diversity among their members. Recent approaches regularize predictions to increase diversity; however, they also drastically decrease individual members' performances. In this paper, we argue that learning strategies for deep ensembles need to tackle the *trade-off between ensemble diversity and individual accuracies*. Motivated by arguments from information theory and leveraging recent advances in neural estimation of conditional mutual information, we introduce a novel training criterion called DICE: it increases diversity by *reducing spurious correlations among features*. The main idea is that features extracted from pairs of members should only share information useful for target class prediction without being conditionally redundant. Therefore, besides the classification loss with information bottleneck, we adversarially prevent features from being conditionally predictable from each other. We manage to reduce simultaneous errors while protecting class information. We obtain state-of-the-art accuracy results on CIFAR-10/100: for example, an ensemble of 5 networks trained with DICE matches an ensemble of 7 networks trained independently. We further analyze the consequences on calibration, uncertainty estimation, out-of-distribution detection and online co-distillation.

## 1 Introduction

Averaging the predictions of several models can significantly improve the generalization ability of a predictive system. Due to its effectiveness, **ensembling** has been a popular research topic (Nilsson, 1965; Hansen & Salamon, 1990; Wolpert, 1992; Krogh & Vedelsby, 1995; Breiman, 1996; Dietterich, 2000; Zhou et al., 2002; Rokach, 2010; Ovadia et al., 2019) as a simple alternative to fully Bayesian methods (Blundell et al., 2015; Gal & Ghahramani, 2016). It is currently the de facto solution for many machine learning applications and Kaggle competitions (Hin, 2020).

Ensembling reduces the variance of estimators (see Appendix E.1) thanks to the diversity in predictions. This reduction is most effective when errors are uncorrelated and members are diverse, *i.e.*, when they do not simultaneously fail on the same examples. Conversely, an ensemble of M identical networks is no better than a single one. In deep ensembles (Lakshminarayanan et al., 2017), **the weights are traditionally trained independently**: diversity among members only relies on the randomness of the initialization and of the learning procedure. Figure 1 shows that the performance of this procedure quickly plateaus with additional members.

To obtain more diverse ensembles, we could adapt the training samples through bagging (Breiman, 1996) and bootstrapping (Efron & Tibshirani, 1994), but a reduction of training samples has a negative impact on members with multiple local minima (Lee et al., 2015). Sequential boosting does not scale well for time-consuming deep learners that overfit their training dataset. Liu & Yao (1999a;b); Brown et al. (2005b) explicitly quantified the diversity and regularized members into having negatively correlated errors. However, these ideas have not significantly improved accuracy when applied to deep learning (Shui et al., 2018; Pang et al., 2019): while members should predict the same target, they force disagreements among strong learners and therefore increase their bias. It highlights the main objective and challenge of our paper: finding a training strategy to reach an improved **trade-off between ensemble diversity and individual accuracies** (Masegosa, 2020).

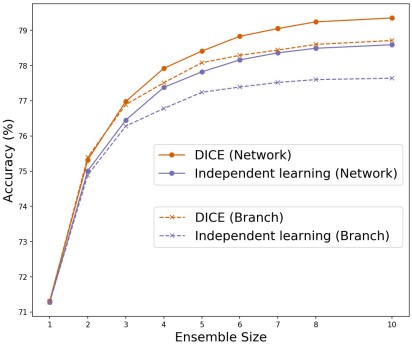

Figure 1: **DICE better leverages ensemble size**. Without weights sharing, 5 networks trained with DICE match 7 networks trained independently. With low-level weights sharing, 4 branches trained with DICE match 7 traditional branches. Dataset: CIFAR-100. Backbone: ResNet-32. Details in Table 8.

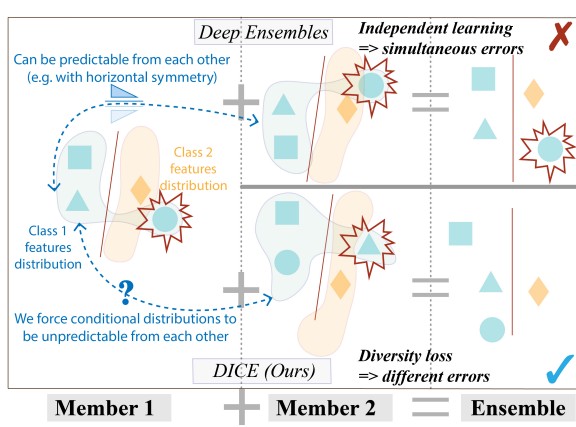

Figure 2: **Outline**. DICE prevents features from being predictable from each other *conditionally* upon the target class. Features extracted by members (1, 2) from one input (●,●) should not share more information than features from two inputs in the same class (●,▲): i.e., (●,-) should not be able to differentiate (-,●) and (-,▲).

Our core approach is to encourage all members **to predict the same thing, but for different reasons**. Therefore the diversity is enforced in the features space and not on predictions. Intuitively, to maximize the impact of a new member, extracted features should bring information about the target that is absent at this time so unpredictable from other members' features. It would remove **spurious correlations**, *e.g.* information redundantly shared among features extracted by different members but useless for class prediction. This redundancy may be caused by a detail in the image background and therefore will not be found in features extracted from other images belonging to the same class. This could make members predict badly simultaneously, as shown in Figure 2.

Our new learning framework, called DICE, is driven by **Information Bottleneck** (IB) (Tishby, 1999; Alemi et al., 2017) principles, that force features to be concise by forgetting the task-irrelevant factors. Specifically, DICE leverages the Minimum Necessary Information criterion (Fischer, 2020) for deep ensembles, and aims at reducing the **mutual information** (MI) between features and inputs, but also information shared between features. We prevent extracted features from being redundant. As mutual information can detect arbitrary dependencies between random variables (such as symmetry, see Figure 2), we increase the distance between pairs of members: it promotes diversity by reducing predictions' covariance. Most importantly, DICE protects features' informativeness by **conditioning** mutual information upon the target. We build upon recent neural approaches (Belghazi et al., 2018) based on the Donsker-Varadhan representation of the KL formulation of MI.

We summarize our contributions as follows:

- We introduce DICE, a new adversarial learning framework to explicitly increase diversity in ensemble by minimizing the conditional redundancy between features.
- We rationalize our training objective by arguments from information theory.
- We propose an implementation through neural estimation of conditional redundancy.

We consistently improve accuracy on CIFAR-10/100 as summarized in Figure 1, with better uncertainty estimation and calibration. We analyze how the two components of our loss modify the accuracy-diversity trade-off. We improve out-of-distribution detection and online co-distillation.

## 2 DICE MODEL

**Notations** Given an input distribution $X$, a network $\theta$ is trained to extract the best possible dense features $Z$ to model the distribution $p_\theta(Y|X)$ over the targets, which should be close to the Dirac on the true label. Our approach is designed for ensembles with $M$ members $\theta_i, i \in \{1, \ldots, M\}$ extracting $Z_i$. In branch-based setup, members share low-level weights to reduce computation cost. We average the $M$ predictions in inference. We initially consider an ensemble of $M = 2$ members.

**Quick overview** First, we train each member separately for classification with information bottleneck. Second, we train members together to remove spurious redundant correlations while training adversarially a discriminator. In conclusion, members learn to classify with conditionally uncorrelated features for increased diversity. Our procedure is driven by the following theoretical findings.

## 2.A DERIVING TRAINING OBJECTIVE

### 2.A.1 BASELINE: NON-CONDITIONAL OBJECTIVE

The Minimum Necessary Information (MNI) criterion from (Fischer, 2020) aims at finding minimal statistics. In deep ensembles, $Z_1$ and $Z_2$ should capture only *minimal* information from $X$, while preserving the *necessary information* about the task $Y$. First, we consider separately the two Markov chains $Z_1 \leftarrow X \leftrightarrow Y$ and $Z_2 \leftarrow X \leftrightarrow Y$. As entropy measures information, entropy of $Z_1$ and $Z_2$ not related to $Y$ should be minimized. We recover IB (Alemi et al., 2017) in deep ensembles: $\text{IB}_{\beta_{ib}}(Z_1, Z_2) = \frac{1}{\beta_{ib}}[I(X; Z_1) + I(X; Z_2)] - [I(Y; Z_1) + I(Y; Z_2)] = \text{IB}_{\beta_{ib}}(Z_1) + \text{IB}_{\beta_{ib}}(Z_2)$. Second, let's consider $I(Z_1; Z_2)$: we minimize it following the minimality constraint of the MNI.

$$\text{IBR}_{\beta_{ib}, \delta_r}(Z_1, Z_2) = \frac{1}{\beta_{ib}} \overbrace{[I(X; Z_1) + I(X; Z_2)]}^{\text{Compression}} - \overbrace{[I(Y; Z_1) + I(Y; Z_2)]}^{\text{Relevancy}} + \delta_r \overbrace{I(Z_1; Z_2)}^{\text{Redundancy}}$$
$$= \text{IB}_{\beta_{ib}}(Z_1) + \text{IB}_{\beta_{ib}}(Z_2) + \delta_r I(Z_1; Z_2).$$

**Analysis** In this baseline criterion, relevancy encourages $Z_1$ and $Z_2$ to capture information about $Y$. Compression & redundancy (R) split the information from $X$ into two compressed & independent views. The relevancy-compression-redundancy trade-off depends on the values of $\beta_{ib}$ & $\delta_r$.

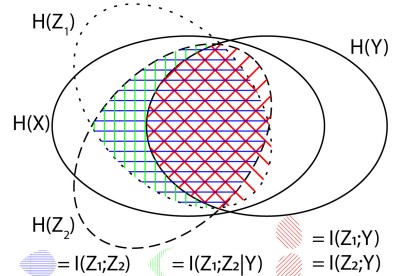

Figure 3: **Venn Information Diagram** (Yeung, 1991). DICE minimizes conditional redundancy (green vertical stripes ▒) with no overlap with relevancy (red stripes).

### 2.A.2 DICE: CONDITIONAL OBJECTIVE

The problem is that the compression and redundancy terms in IBR also reduce necessary information related to $Y$: it is detrimental to have $Z_1$ and $Z_2$ fully disentangled while training them to predict the same $Y$. As shown on Figure 3, redundancy regions (blue horizontal stripes ▤) overlap with relevancy regions (red stripes). Indeed, the true constraints that the MNI criterion really entails are the following **conditional** equalities given $Y$:

$$I(X; Z_1|Y) = I(X; Z_2|Y) = I(Z_1; Z_2|Y) = 0.$$

Mutual information being non-negative, we transform them into our main DICE objective:

$$\begin{aligned}
&\text{DICE}_{\beta_{ceb}, \delta_{cr}}(Z_1, Z_2) \\
&= \frac{1}{\beta_{ceb}} \underbrace{[I(X; Z_1|Y) + I(X; Z_2|Y)]}_{\text{Conditional Compression}} - \underbrace{[I(Y; Z_1) + I(Y; Z_2)]}_{\text{Relevancy}} + \delta_{cr} \underbrace{I(Z_1; Z_2|Y)}_{\text{Conditional Redundancy}} \\
&= \text{CEB}_{\beta_{ceb}}(Z_1) + \text{CEB}_{\beta_{ceb}}(Z_2) + \delta_{cr} I(Z_1; Z_2|Y),
\end{aligned} \quad (1)$$

where we recover two conditional entropy bottleneck (CEB) (Fischer, 2020) components, $\text{CEB}_{\beta_{ceb}}(Z_i) = \frac{1}{\beta_{ceb}} I(X; Z_i|Y) - I(Y; Z_i)$, with $\beta_{ceb} > 0$ and $\delta_{cr} > 0$.

**Analysis** The relevancy terms force features to be informative about the task $Y$. But contrary to IBR, DICE bottleneck constraints only minimize irrelevant information to $Y$. First, the conditional compression removes in $Z_1$ (or $Z_2$) information from $X$ not relevant to $Y$. Second, the conditional redundancy (CR) reduces spurious correlations between members and only forces them to **have independent bias, but definitely not independent features**. It encourages diversity without affecting members' individual precision as it protects information related to the target class in $Z_1$ and $Z_2$. Useless information from $X$ to predict Y should certainly not be in $Z_1$ or $Z_2$, but it is even worse if they are in $Z_1$ *and* $Z_2$ simultaneously as it would cause simultaneous errors. Even if for $i \in \{1, 2\}$, reducing $I(Z_i, X|Y)$ indirectly controls $I(Z_1, Z_2|Y)$ (as $I(Z_1; Z_2|Y) \leq I(X; Z_i|Y)$ by chain rule), it is more efficient to directly target this intersection region through the CR term. In a final word, DICE is to IBR for deep ensembles as CEB is to IB for a single network.

We now approximate the two CEB and the CR components in DICE objective from equation 1.

## 2.B Approximating DICE into a Tractable Loss

### 2.B.1 Variational Approximation of Conditional Entropy Bottleneck

We leverage Markov assumptions in $Z_i \leftarrow X \leftrightarrow Y, i \in \{1, 2\}$ and empirically estimate on the classification training dataset of $N$ i.i.d. points $D = \{x^n, y^n\}_{n=1}^{N}, y^n \in \{1, \ldots, K\}$. Following Fischer (2020), $CEB_{\beta_{ceb}}(Z_i) = \frac{1}{\beta_{ceb}} I(X; Z_i|Y) - I(Y; Z_i)$ is variationally upper bounded by:

$$\text{VCEB}_{\beta_{ceb}}(\{e_i, b_i, c_i\}) = \frac{1}{N} \sum_{n=1}^{N} \frac{1}{\beta_{ceb}} D_{\text{KL}} \left(e_i(z|x^n) \| b_i(z|y^n)\right) - \mathbb{E}_\epsilon \left[\log c_i(y^n|e_i(x^n, \epsilon))\right]. \quad (2)$$

See explanation in Appendix E.4. $e_i(z|x)$ is the true features distribution generated by the *encoder*, $c_i(y|z)$ is a variational approximation of true distribution $p(y|z)$ by the *classifier*, and $b_i(z|y)$ is a variational approximation of true distribution $p(z|y)$ by the *backward* encoder. This loss is applied separately on each member $\theta_i = \{e_i, c_i, b_i\}, i \in \{1, 2\}$.

Practically, we parameterize all distributions with Gaussians. The encoder $e_i$ is a traditional neural network features extractor (*e.g.* ResNet-32) that learns **distributions** (means and covariances) rather than deterministic points in the features space. That's why $e_i$ transforms an image into 2 tensors; a features-mean $e_i^\mu(x)$ and a diagonal features-covariance $e_i^\sigma(x)$ each of size $d$ (*e.g.* 64). The classifier $c_i$ is a dense layer that transforms a features-sample $z$ into logits to be aligned with the target $y$ through conditional cross entropy. $z$ is obtained via reparameterization trick: $z = e_i(x, \epsilon) = e_i^\mu(x) + \epsilon e_i^\sigma(x)$ with $\epsilon \sim N(0, 1)$. Finally, the backward encoder $b_i$ is implemented as an embedding layer of size $(K, d)$ that maps the $K$ classes to class-features-means $b_i^\mu(z|y)$ of size $d$, as we set the class-features-covariance to $\mathbf{1}$. The Gaussian parametrization also enables the exact computation of the $D_{\text{KL}}$ (see Appendix E.3), that forces (1) features-mean $e_i^\mu(x)$ to converge to the class-features-mean $b_i^\mu(z|y)$ and (2) the predicted features-covariance $e_i^\sigma(x)$ to be close to $\mathbf{1}$. The **advantage of VCEB versus VIB** (Alemi et al., 2017) is the **class conditional** $b_i^\mu(z|y)$ versus non-conditional $b_i^\mu(z)$ which protects class information.

### 2.B.2 Adversarial Estimation of Conditional Redundancy

**Theoretical Problem** We now focus on estimating $I(Z_1; Z_2|Y)$, with no such Markov properties. Despite being a pivotal measure, mutual information estimation historically relied on nearest neighbors (Singh et al., 2003; Kraskov et al., 2004; Gao et al., 2018) or density kernels (Kandasamy et al., 2015) that do not scale well in high dimensions. We benefit from recent advances in **neural estimation of mutual information** (Belghazi et al., 2018), built on optimizing Donsker & Varadhan (1975) dual representations of the KL divergence. Mukherjee et al. (2020) extended this formulation for conditional mutual information estimation.

$$\begin{aligned}
\text{CR} = I(Z_1; Z_2|Y) &= D_{\text{KL}}(P(Z_1, Z_2, Y) \| P(Z_1, Y) p(Z_2|Y)) \\
&= \sup_f \mathbb{E}_{x \sim p(z_1, z_2, y)}[f(x)] - \log \left(\mathbb{E}_{x \sim p(z_1, y) p(z_2|y)}[\exp(f(x))]\right) \\
&= \mathbb{E}_{x \sim p(z_1, z_2, y)}[f^*(x)] - \log \left(\mathbb{E}_{x \sim p(z_1, y) p(z_2|y)}[\exp(f^*(x))]\right),
\end{aligned}$$

where $f^*$ computes the pointwise likelihood ratio, *i.e.*, $f^*(z_1, z_2, y) = \frac{p(z_1, z_2, y)}{p(z_1, y) p(z_2|y)}$.

**Empirical Neural Estimation** We estimate CR (1) using the empirical data distribution and (2) replacing $f^* = \frac{w^*}{1 - w^*}$ by the output of a discriminator $w$, trained to imitate the optimal $w^*$. Let $\mathcal{B}_\text{J}$ be a batch sampled from the observed joint distribution $p(z_1, z_2, y) = p(e_1(z|x), e_2(z|x), y)$; we select the features extracted by the two members from one input. Let $\mathcal{B}_p$ be sampled from the product distribution $p(z_1, y) p(z_2|y) = p(e_1(z|x), y) p(z_2|y)$; we select the features extracted by the two members from two different inputs that share the same class. We train a multi-layer network $w$ on the binary task of distinguishing these two distributions with the standard cross-entropy loss:

$$\mathcal{L}_{ce}(w) = -\frac{1}{|\mathcal{B}_\text{J}| + |\mathcal{B}_p|} \left[ \sum_{(z_1, z_2, y) \in \mathcal{B}_\text{J}} \log w(z_1, z_2, y) + \sum_{(z_1, z_2', y) \in \mathcal{B}_p} \log(1 - w(z_1, z_2', y)) \right]. \quad (3)$$

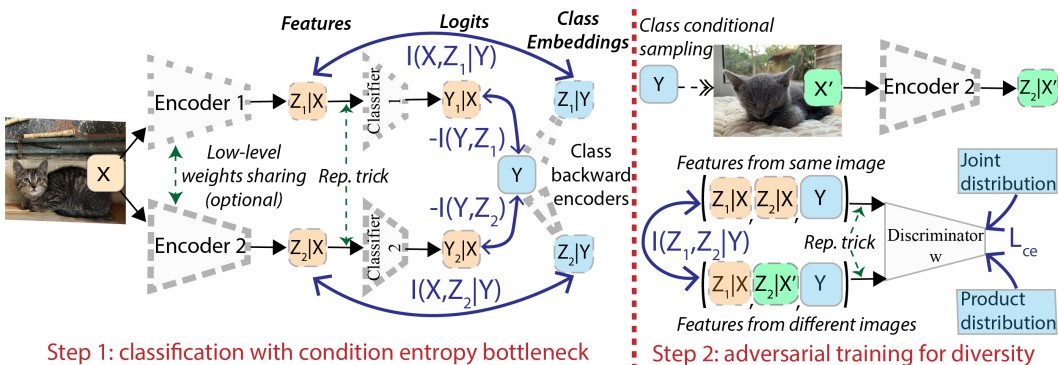

**Figure 4: Learning strategy overview**. Blue arrows represent training criteria: (1) classification with conditional entropy bottleneck applied separately on members 1 and 2, and (2) adversarial training to delete spurious correlations between members and increase diversity. $X$ and $X'$ belong to the same $Y$ for **conditional redundancy** minimization. See Figure 13 for a larger version.

If $w$ is calibrated (see Appendix B.3), a consistent (Mukherjee et al., 2020) estimate of CR is:

$$\hat{\mathcal{I}}_{DV}^{CR} = \frac{1}{|\mathcal{B}_{\text{J}}|} \sum_{(z_1, z_2, y) \in \mathcal{B}_{\text{J}}} \underbrace{\log f(z_1, z_2, y)}_{\text{Diversity}} - \log \left( \frac{1}{|\mathcal{B}_p|} \sum_{(z_1, z_2', y) \in \mathcal{B}_p} \underbrace{f(z_1, z_2', y)}_{\text{Fake correlations}} \right), \text{ with } f = \frac{w}{1-w}.$$

**Intuition** By training our members to minimize $\hat{\mathcal{I}}_{DV}^{CR}$, we force triples from the joint distribution to be indistinguishable from triples from the product distribution. Let's imagine that two features are conditionally correlated, some spurious information is shared between features only when they are from the same input and not from two inputs (from the same class). This correlation can be informative about a detail in the background, an unexpected shape in the image, that is rarely found in samples from this input's class. In that case, the product and joint distributions are easily distinguishable by the discriminator. The first adversarial component will force the extracted features to reduce the correlation, and ideally one of the two features loses this information: it reduces redundancy and increases diversity. The second term would create fake correlations between features from different inputs. As we are not interested in a precise estimation of the CR, we get rid of this second term that, empirically, did not increase diversity, as detailed in Appendix G.

$$\hat{\mathcal{L}}_{DV}^{CR}(e_1, e_2) = \frac{1}{|\mathcal{B}_{\text{J}}|} \sum_{(z_1, z_2, y) \in \mathcal{B}_{\text{J}} \sim p(e_1(z|x), e_2(z|x)), y)} \log f(z_1, z_2, y). \tag{4}$$

**Summary** First, we train each member for classification with VCEB from equation 2, as shown in Step 1 from Figure 4. Second, as shown in Step 2 from Figure 4, the discriminator, conditioned on the class $Y$, learns to distinguish features sampled from one image versus features sampled from two images belonging to $Y$. Simultaneously, both members adversarially (Goodfellow et al., 2014) delete spurious correlations to reduce CR estimation from equation 4 with differentiable signals: it conditionally aligns features. We provide a pseudo-code in B.4. While we derive similar losses for IBR and CEBR in Appendix E.5, the full DICE loss is finally:

$$\mathcal{L}_{DICE}(\theta_1, \theta_2) = \text{VCEB}_{\beta_{ceb}}(\theta_1) + \text{VCEB}_{\beta_{ceb}}(\theta_2) + \delta_{cr}\hat{\mathcal{L}}_{DV}^{CR}(e_1, e_2). \tag{5}$$

## 2.C FULL PROCEDURE WITH M MEMBERS

We expand our objective for an ensemble with $M > 2$ members. We only consider pairwise interactions for simplicity to keep quadratic rather than exponential growth in number of components and truncate higher order interactions, *e.g.* $I(Z_i; Z_j, Z_k|Y)$ (see Appendix F.1). Driven by previous variational and neural estimations, we train $\theta_i = \{e_i, b_i, c_i\}, i \in \{1, \ldots, M\}$ on:

$$\mathcal{L}_{DICE}(\theta_{1:M}) = \sum_{i=1}^{M} \text{VCEB}_{\beta_{ceb}}(\theta_i) + \frac{\delta_{cr}}{(M-1)} \sum_{i=1}^{M} \sum_{j=i+1}^{M} \hat{\mathcal{L}}_{DV}^{CR}(e_i, e_j), \tag{6}$$

while training **adversarially** $w$ on $\mathcal{L}_{ce}$. Batch $\mathcal{B}_{j}$ is sampled from the concatenation of joint distribution $p(z_i, z_j, y)$ where $i, j \in \{1, \dots, M\}, i \neq j$, while $\mathcal{B}_p$ is sampled from the product distribution, $p(z_i, y)p(z_j|y)$. We use the same discriminator $w$ for $\binom{M}{2}$ estimates. It improves scalability by reducing the number of parameters to be learned. Indeed, an additional member in the ensemble only adds $256 * d$ trainable weights in $w$, where $d$ is the features dimension. See Appendix B.3 for additional information related to the discriminator $w$.

## 3 RELATED WORK

To reduce the training cost of deep ensembles (Hansen & Salamon, 1990; Lakshminarayanan et al., 2017), Huang et al. (2017) collect **snapshots** on training trajectories. One stage end-to-end **co-distillation** (Song & Chai, 2018; Lan et al., 2018; Chen et al., 2020b) share low-level features among members in *branch-based* ensemble while forcing each member to mimic a dynamic weighted combination of the predictions to increase individual accuracy. However both methods correlate errors among members, homogenize predictions and fail to fit the different modes of the data which overall reduce diversity.

Beyond random initializations (Kolen & Pollack, 1991), authors **implicitly** introduced stochasticity into the training, by providing *subsets of data* to learners with bagging (Breiman, 1996) or by backpropagating *subsets of gradients* (Lee et al., 2016); however, the reduction of training samples hurts performance for sufficiently complex models that overfit their training dataset (Nakkiran et al., 2019). Boosting with sequential training is not suitable for deep members (Lakshminarayanan et al., 2017). Some approaches applied different *data augmentations* (Dvornik et al., 2019; Stickland & Murray, 2020), used *different networks* or *hyperparameters* (Singh et al., 2016; Ruiz & Verbeek, 2020; Yang & Soatto, 2020), but are not general-purpose and depend on specific engineering choices.

Others **explicitly** encourage orthogonality of the *gradients* (Ross et al., 2020; Kariyappa & Qureshi, 2019; Dabouei et al., 2020) or of the *predictions*, by *boosting* (Freund & Schapire, 1999; Margineantu & Dietterich) or with a *negative correlation* regularization (Shui et al., 2018), but they reduce members accuracy. Second-order PAC-Bayes bounds motivated the diversity loss in Masegosa (2020). As far as we know, adaptive diversity promoting (ADP) (Pang et al., 2019) is the unique approach more accurate than the independent baseline: they decorrelate the non-maximal predictions. The limited success of these logits approaches suggests that we seek diversity in *features*. Empirically we found that the increase of $(L_1, L_2, -\cos)$ distances between features (Kim et al., 2018) reduce performance: they are not invariant to variables' symmetry. Simultaneously to our findings, Sinha et al. (2020) is somehow equivalent to our IBR objective (see Appendix C.2) but without information bottleneck motivations for the diversity loss.

The uniqueness of **mutual information** (see Appendix E.2) as a distance measure between variables has been applied in countless machine learning projects, such as reinforcement learning (Kim et al., 2019a), metric learning (Kemertas et al., 2020), or evolutionary algorithms (Aguirre & Coello, 2004). Objectives are often a trade-off between (1) informativeness and (2) compression. In computer vision, unsupervised deep representation learning (Hjelm et al., 2019; van den Oord et al., 2018; Tian et al., 2020a; Bachman et al., 2019) maximizes correlation between features and inputs following Infomax (Linsker, 1988; Bell & Sejnowski, 1995), while discarding information not shared among different views (Bhardwaj et al., 2020), or penalizing predictability of one latent dimension given the others for disentanglement (Schmidhuber, 1992; Comon, 1994; Kingma & Welling, 2014; Kim & Mnih, 2018; Blot et al., 2018).

The ideal level of compression is **task dependent** (Soatto & Chiuso, 2014). As a selection criterion, features should not be redundant (Battiti, 1994; Peng et al., 2005) but relevant and complementary given the task (Novovičová et al., 2007; Brown, 2009). As a learning criteria, correlations between features and inputs are minimized according to Information Bottleneck (Tishby, 1999; Alemi et al., 2017; Kirsch et al., 2020; Saporta et al., 2019), while those between features and targets are maximized (LeCun et al., 2006; Qin & Kim, 2019). It forces the features to ignore task-irrelevant factors (Zhao et al., 2020), to reduce overfitting (Alemi et al., 2018) while protecting needed information (Tian et al., 2020b). Fischer & Alemi (2020) concludes in the superiority of conditional alignment to reach the MNI point.

## 4 EXPERIMENTS

In this section, we present our experimental results on the *CIFAR-10* and *CIFAR-100* (Krizhevsky et al., 2009) datasets. We detail our implementation in Appendix B. We took most hyperparameter values from Chen et al. (2020b). Hyperparameters for adversarial training and information bottle-neck were fine-tuned on a validation dataset made of 5% of the training dataset, see Appendix D.1. **Bold** highlights best score. First, we show gain in accuracy. Then, we further analyze our strategy's impacts on calibration, uncertainty estimation, out-of-distribution detection and co-distillation.

### 4.A COMPARISON OF CLASSIFICATION ACCURACY

Table 1: **CIFAR-100 ensemble classification accuracy** (Top-1, %).

| Name | Components | | ResNet-32 | | | | ResNet-110 | | WRN-28-2 | | 3-net |
| :---: | :---: | :---: | :---: | :---: | :---: | :---: | :---: | :---: | :---: | :---: | :---: |
| | Div. | I.B. | 3-branch | 4-branch | 5-branch | 4-net | 3-branch | 4-branch | 3-branch | 4-branch | 3-net |
| Ind. | | | $76.28_{\pm0.12}$ | $76.78_{\pm0.19}$ | $77.24_{\pm0.25}$ | $77.38_{\pm0.12}$ | $80.54_{\pm0.09}$ | $80.89_{\pm0.31}$ | $78.83_{\pm0.12}$ | $79.10_{\pm0.08}$ | $80.01_{\pm0.15}$ |
| ONE (Lan et al., 2018) | | | $75.17_{\pm0.35}$ | $75.13_{\pm0.25}$ | $75.25_{\pm0.22}$ | $76.25_{\pm0.32}$ | $78.97_{\pm0.24}$ | $79.86_{\pm0.25}$ | $78.38_{\pm0.45}$ | $78.47_{\pm0.32}$ | $77.53_{\pm0.36}$ |
| OKDDip (Chen et al., 2020b) | | | $75.37_{\pm0.32}$ | $76.85_{\pm0.25}$ | $76.95_{\pm0.18}$ | $77.27_{\pm0.31}$ | $79.07_{\pm0.27}$ | $80.46_{\pm0.35}$ | $79.01_{\pm0.19}$ | $79.32_{\pm0.17}$ | $80.02_{\pm0.14}$ |
| ADP (Pang et al., 2019) | Pred. | | $76.37_{\pm0.11}$ | $77.21_{\pm0.21}$ | $77.67_{\pm0.25}$ | $77.51_{\pm0.25}$ | $80.73_{\pm0.38}$ | $81.40_{\pm0.27}$ | $79.21_{\pm0.19}$ | $79.71_{\pm0.18}$ | $80.01_{\pm0.17}$ |
| IB (equation 8) | | VIB | $76.01_{\pm0.12}$ | $76.93_{\pm0.24}$ | $77.22_{\pm0.19}$ | $77.72_{\pm0.12}$ | $80.43_{\pm0.34}$ | $81.12_{\pm0.19}$ | $79.19_{\pm0.16}$ | $79.15_{\pm0.12}$ | $80.15_{\pm0.13}$ |
| CEB (equation 2) | | VCEB | $76.36_{\pm0.06}$ | $76.98_{\pm0.18}$ | $77.35_{\pm0.14}$ | $77.64_{\pm0.15}$ | $81.08_{\pm0.12}$ | $81.17_{\pm0.16}$ | $78.92_{\pm0.08}$ | $79.20_{\pm0.13}$ | $80.38_{\pm0.18}$ |
| IBR (equation 9) | R | VIB | $76.68_{\pm0.13}$ | $77.25_{\pm0.13}$ | $77.77_{\pm0.21}$ | $77.84_{\pm0.12}$ | $81.34_{\pm0.21}$ | $81.38_{\pm0.08}$ | $79.33_{\pm0.15}$ | $79.90_{\pm0.10}$ | $80.22_{\pm0.10}$ |
| CEBR (equation 10) | R | VCEB | $76.72_{\pm0.08}$ | $77.30_{\pm0.12}$ | $77.81_{\pm0.10}$ | $77.82_{\pm0.11}$ | $81.52_{\pm0.11}$ | $81.55_{\pm0.33}$ | $79.25_{\pm0.15}$ | $79.98_{\pm0.07}$ | $80.35_{\pm0.15}$ |
| DICE (equation 6) | CR | VCEB | $\mathbf{76.89}_{\pm0.09}$ | $\mathbf{77.51}_{\pm0.17}$ | $\mathbf{78.08}_{\pm0.18}$ | $\mathbf{77.92}_{\pm0.08}$ | $\mathbf{81.67}_{\pm0.14}$ | $\mathbf{81.93}_{\pm0.13}$ | $\mathbf{79.59}_{\pm0.13}$ | $\mathbf{80.05}_{\pm0.11}$ | $\mathbf{80.55}_{\pm0.12}$ |

Table 1 reports the Top-1 classification accuracy averaged over 3 runs with standard deviation for CIFAR-100, while Table 2 focuses on CIFAR-10. {3,4,5}-{branch,net} refers to the training of {3,4,5} members {with,without} low-level weights sharing. Ind. refers to *independent* deterministic deep ensembles without interactions between members (except optionally the low-level weights sharing). DICE surpasses concurrent approaches (summarized in Appendix C) for ResNet and Wide-ResNet architectures, in network and even more in branch setup. We bring significant and systematic improvements to the current state-of-the-art ADP (Pang et al., 2019): e.g., {+0.52, +0.30, +0.41} for {3,4,5}-branches ResNet-32, {+0.94, +0.53} for {3,4}-branches ResNet-110 and finally +0.34 for 3-networks WRN-28-2. Diversity approaches better leverage size, as shown on the main Figure 1, which is detailed in Table 8: on CIFAR-100, DICE outperforms Ind. by {+0.60, +0.73, +0.84} for {3,4,5}-branches ResNet-32. Finally, learning only the redundancy loss without compression yields unstable results: CEB learns a distribution (at almost no extra cost) that stabilizes adversarial training (see Appendix F.1) through sampling, with lower standard deviation in results than IB ($\beta_{ib}$ can hinder the learnability (Wu et al., 2019b)).

Table 2: **CIFAR-10 ensemble classification accuracy** (Top-1, %).

| Backbone | Structure | Ind. | ONE | OKDDip | ADP | IB | CEB | IBR | CEBR | DICE |
| :---: | :---: | :---: | :---: | :---: | :---: | :---: | :---: | :---: | :---: | :---: |
| ResNet-32 | 4-branch | $94.75_{\pm0.08}$ | $94.41_{\pm0.05}$ | $94.86_{\pm0.08}$ | $94.92_{\pm0.04}$ | $94.76_{\pm0.12}$ | $94.93_{\pm0.11}$ | $94.91_{\pm0.14}$ | $94.94_{\pm0.12}$ | $\mathbf{95.01}_{\pm0.09}$ |
| ResNet-110 | 3-branch | $95.62_{\pm0.06}$ | $95.25_{\pm0.08}$ | $95.21_{\pm0.09}$ | $95.43_{\pm0.12}$ | $94.54_{\pm0.07}$ | $94.65_{\pm0.05}$ | $95.68_{\pm0.05}$ | $95.67_{\pm0.06}$ | $\mathbf{95.74}_{\pm0.08}$ |

### 4.B ABLATION STUDY

Branch-based is attractive: it reduces bias by gradient diffusion among shared layers, at only a slight cost in diversity which makes our approach even more valuable. We therefore study the 4-branches ResNet-32 on CIFAR-100 in following experiments. We ablate the two components of DICE: (1) deterministic, with VIB or VCEB, and (2) no adversarial loss, or with redundancy, conditionally or not. We measure diversity by the ratio-error (Aksela, 2003), $r = \frac{N_{\text{single}}}{N_{\text{shared}}}$, which computes the ratio between the number of single errors $N_{\text{single}}$ and of shared errors $N_{\text{shared}}$. A higher average over the $\binom{M}{2}$ pairs means higher diversity as members are less likely to err on the same inputs. Our analysis remains valid for non-pairwise diversity measures, analyzed in Appendix A.5.

In Figure 5, CEB has slightly higher diversity than Ind.: it benefits from compression. ADP reaches higher diversity but sacrifices individual accuracies. On the contrary, co-distillation OKDDip sacri-

fices diversity for individual accuracies. DICE curve is above all others, and notably $\delta_{cr} = 0.2$ induces an **optimal trade-off between ensemble diversity and individual accuracies** on validation. CEBR reaches same diversity with lower individual accuracies: information about $Y$ is removed.

Figure 6 shows that starting from random initializations, diversity begins small: DICE minimizes the estimated CR in features and increases diversity in predictions compared to CEB ($\delta_{cr} = 0.0$). The effect is correlated with $\delta_{cr}$: a high value (0.6) creates too much diversity. On the contrary, a negative value ($-0.025$) can decrease diversity. Figure 8 highlights opposing dynamics in accuracies.

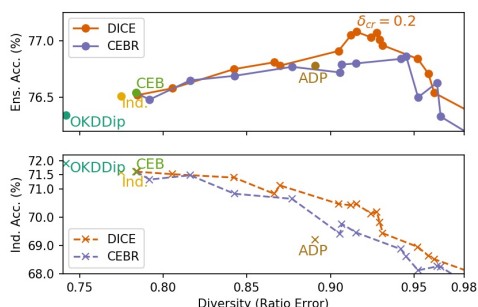

Figure 5: **Ensemble diversity/individual accuracy trade-off** for different strategies. DICE (r. CEBR) is learned with different $\delta_{cr}$ (r. $\delta_r$).

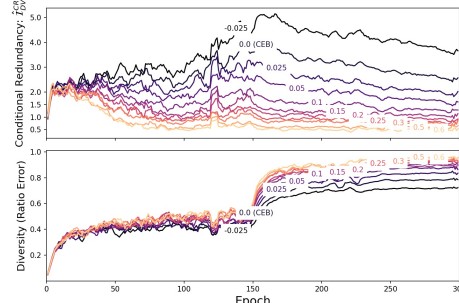

Figure 6: **Impact of the diversity coefficient** $\delta_{cr}$ in DICE on the training dynamics on validation: CR is negatively correlated with diversity.

### 4.c FURTHER ANALYSIS: UNCERTAINTY ESTIMATION AND CALIBRATION

**Procedure** We follow the procedure from (Ashukha et al., 2019). To evaluate the quality of the uncertainty estimates, we reported two complementary proper scoring rules (Gneiting & Raftery, 2007); the *Negative Log-Likelihood* (NLL) and the *Brier Score* (BS) (Brier, 1950). To measure the calibration, *i.e.*, how classification confidences match the observed prediction accuracy, we report the *Expected Calibration Error* (ECE) (Naeini et al., 2015) and the *Thresholded Adaptive Calibration Error* (TACE) (Nixon et al., 2019) with 15 bins: TACE resolves some pathologies in ECE by thresholding and adaptive binning. Ashukha et al. (2019) showed that "comparison of [. . . ] ensembling methods without temperature scaling (Guo et al., 2017) might not provide a fair ranking". Therefore, we randomly divide the test set into two equal parts and compute metrics for each half using the temperature $T$ optimized on another half: their mean is reported. Table 3 compares results after temperature scaling (TS) while those before TS are reported in Table 9 in Appendix A.6.

Table 3: **Uncertainty estimation** (NLL, BS) and **calibration** (ECE, TACE) on CIFAR-100 **after** temperature scaling.

| | 1-net | Ind. | OKDDip-E | ADP | IB | CEB | IBR | CEBR | DICE |
|---|---|---|---|---|---|---|---|---|---|
| $T$ | 1.49 | 1.31 | 1.33 | 0.64 | 1.21 | 1.24 | 1.17 | 1.19 | 1.11 |
| NLL $\downarrow (10^{-1})$ | 10.38 | 8.10 | 8.13 | 8.51 | 8.12 | 8.11 | 8.09 | 8.05 | **7.98** |
| BS $\downarrow (10^{-3})$ | 3.92 | 3.24 | 3.19 | 3.27 | 3.20 | 3.19 | 3.17 | 3.18 | **3.12** |
| ECE $\downarrow (10^{-2})$ | 1.83 | **1.60** | 1.73 | 2.99 | 2.17 | 2.07 | 1.97 | 2.02 | 2.59 |
| TACE $\downarrow (10^{-3})$ | 1.98 | 1.78 | 1.74 | 1.79 | **1.68** | 1.69 | 1.75 | 1.72 | 1.70 |
| Acc. $\uparrow (\%)$ | 71.28 | 76.71 | 76.85 | 77.21 | 76.93 | 76.98 | 77.25 | 77.30 | **77.51** |

**Results** We recover that ensembling improves performances (Ovadia et al., 2019), as one single network (1-net) performs significantly worse than ensemble approaches with 4-branches ResNet-32. Members' disagreements decrease internal temperature and increase uncertainty estimation. DICE performs best even after TS, and reduces NLL from 8.13 to 7.98 and BS from 3.24 to 3.12 compared to independant learning. Calibration criteria benefit from diversity though they do "not provide a consistent ranking" as stated in Ashukha et al. (2019): for example, we notice that ECE highly depends on hyperparameters, especially $\delta_{cr}$, as shown on Figure 8 in Appendix A.4.

## 4.D Further Analysis: Discriminator Behaviour through OOD Detection

To measure the ability of our ensemble to distinguish in- and out-of-distribution (OOD) images, we consider other datasets at test time following (Hendrycks & Gimpel, 2017) (see Appendix D.2). The confidence score is estimated with the maximum softmax value: the confidence for OOD images should ideally be lower than for CIFAR-100 test images.

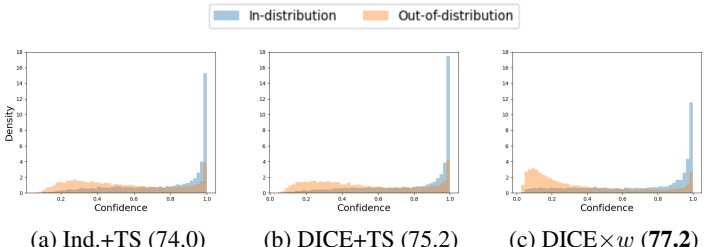

(a) Ind.+TS (74.0)  (b) DICE+TS (75.2)  (c) DICE×$w$ (**77.2**)

Figure 7: **Confidence estimates** separate images from CIFAR-100 and OOD images from TinyImageNet (crop) for different strategies (AUROC ↑). DICE×$w$ uses the discriminator to scale its confidence: $1 - w$'s predictions behave like an "input-dependant temperature".

Temperature scaling (results in Table 7) refines performances (results without TS in Table 6). DICE beats Ind. and CEB in both cases. Moreover, we suspected that features were more correlated for OOD images: they may share redundant artifacts. DICE×$w$ multiplies the classification logits by the mean over all pairs of $1 - w(z_i, z_j, \hat{y}), i \neq j$, with predicted $\hat{y}$ (as the true $y$ is not available at test time). DICE×$w$ performs even better than DICE+TS, but at the cost of additional operations. It shows that $w$ can detect spurious correlations, adversarially deleted only when found in training.

## 4.E Further Analysis: Diverse Teacher for Improved Co-distillation

Table 4: **Individual accuracy for branch-based co-distillation** on CIFAR-100

| | | 1-net | Ind. | ONE (Lan et al., 2018) | OKDDip (Chen et al., 2020b) | | | PCL (Wu & Gong, 2020) | OKDDip+CEB | | | OKDDip+DICE | | |
|---|---|---|---|---|---|---|---|---|---|---|---|---|---|---|
| $T$ co-distillation | | - | - | 3 | 3 | 2.5 | 2 | 3 | 3 | 2.5 | 2 | 3 | 2.5 | 2 |
| ResNet-32 | 3-branch | 71.28±0.11 | 72.15±0.08 | 73.32±0.22 | 73.90±0.15 | 74.01±0.08 | 74.12±0.12 | 74.14±0.16 | 73.95±0.09 | 74.10±0.09 | 74.08±0.11 | 74.14±0.11 | 74.28±0.12 | **74.56**±0.18 |
| | 4-branch | | | 73.42±0.18 | 74.40±0.13 | 74.42±0.11 | 74.31±0.09 | - | 74.01±0.11 | 74.15±0.21 | 74.61±0.17 | 74.22±0.08 | 74.43±0.18 | **74.95**±0.15 |

The inference time in network-ensembles grows linearly with M. Sharing early-features is one solution. We experiment another one by using only the M-th branch at test time. We combine DICE with OKDDip (Chen et al., 2020b): the M-th branch (= the student) learns to mimic the soft predictions from the M-1 first branches (= the teacher), among which we enforce diversity. Our teacher has lower internal temperature (as shown in Experiment 4.c): DICE performs best when soft predictions are generated with lower $T$. We improve state-of-the-art by $\{+0.42, +0.53\}$ for $\{3,4\}$-branches.

## 5 Conclusion

In this paper, we addressed the task of improving deep ensembles' learning strategies. Motivated by arguments from information theory, we derive a novel adversarial diversity loss, based on conditional mutual information. We tackle the trade-off between individual accuracies and ensemble diversity by deleting spurious and redundant correlations. We reach state-of-the-art performance on standard image classification benchmarks. In Appendix F.2, we also show how to regularize deterministic encoders with conditional redundancy without compression: this increases the applicability of our research findings. The success of many real-world systems in production depends on the robustness of deep ensembles: we hope to pave the way towards general-purpose strategies that go beyond independent learning.

### Acknowledgments

This work was granted access to the HPC resources of IDRIS under the allocation 20XX-AD011011953 made by GENCI. We acknowledge the financial support by the ANR agency in the chair VISA-DEEP (project number ANR-20-CHIA-0022-01). Finally, we would like to thank those who helped and supported us during these confinements, in particular Julie and Rouille.

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

# Appendices

Appendix A shows additional experiments. Appendix B describes our implementation to facilitate reproduction. In Appendix C, we summarize the concurrent approaches (see Table 10). In Appendix D, we describe the datasets and the metrics used in our experiments. Appendix E clarifies certain theoretical formulations. In Appendix F, we explain that DICE is a second-order approximation in terms of information interactions and then we try to apply our diversity regularization to deterministic encoders. Appendix G motivates the removal of the second term from our neural estimation of conditional redundancy. We conclude with a sociological analogy in Appendix H.

## A    ADDITIONAL EXPERIMENTS

### A.1    COMPARISONS WITH CO-DISTILLATION AND SNAPSHOT-BASED APPROACHES

Table 5: **Ensemble Accuracy** on different setups. Concurrent approaches' accuracies are those reported in recent papers. DICE outperforms co-distillation and snapshot-based ensembles collected on the training trajectory, which fail to capture the different modes of the data (Ashukha et al., 2019).

| Dataset | Architecture | | | Concurrent Approach | | | Baseline | Ours |
|---|---|---|---|---|---|---|---|---|
| Dataset | Backbone | Structure | Ens. Size | Name | Acc. | According to | Ind. Acc. | DICE Acc. |
| CIFAR-100 | ResNet-32 | Branches | 3 | CL-ILR (Song & Chai, 2018) | 72.99 | (Chen et al., 2020b) | 76.28 | **76.89** |
| | | Nets | 3 | DML (Zhang et al., 2018) | 76.11 | (Chung et al., 2020) | 76.45 | **76.98** |
| | | | | AFD (Chung et al., 2020) | 76.64 | (Chung et al., 2020) | | |
| | ResNet-110 | Branches | 3 | FFL (Kim et al., 2019b) | 78.22 | (Wu & Gong, 2020) | 80.54 | **81.67** |
| | | | | PCL-E (Wu & Gong, 2020) | 80.51 | (Wu & Gong, 2020) | | |
| | | | 4 | CL-ILR (Song & Chai, 2018) | 79.81 | (Chen et al., 2020b) | 80.89 | **81.93** |
| | | Nets | 5 | SWAG (Maddox et al., 2019) | 77.69 | (Ashukha et al., 2019) | 81.7 (Ashukha et al., 2019) | **81.82** |
| | | | | Cyclic SGLD (Zhang et al., 2019) | 74.27 | (Ashukha et al., 2019) | | |
| | | | | Fast Geometric Ens (Garipov et al., 2018) | 78.78 | (Ashukha et al., 2019) | | |
| | | | | Variational Inf. (FFG) (Wu et al., 2019a) | 77.59 | (Ashukha et al., 2019) | | |
| | | | | KFAC-Laplace (Ritter et al., 2018) | 77.13 | (Ashukha et al., 2019) | | |
| | | | | Snapshot Ensembles (Huang et al., 2017) | 77.17 | (Ashukha et al., 2019) | | |
| | WRN-28-2 | Nets | 3 | DML (Zhang et al., 2018) | 79.41 | (Chung et al., 2020) | 80.01 | **80.55** |
| | | | | AFD (Chung et al., 2020) | 79.78 | (Chung et al., 2020) | | |
| CIFAR-10 | ResNet-110 | Branches | 3 | FFL (Kim et al., 2019b) | 95.01 | (Wu & Gong, 2020) | 95.62 | **95.74** |
| | | | | PCL-E (Wu & Gong, 2020) | 95.58 | (Wu & Gong, 2020) | | |

### A.2    OUT-OF-DISTRIBUTION DETECTION

Table 6 summarizes our OOD experiments in the 4-branches ResNet-32 setup. We recover that IB improves OOD detection (Alemi et al., 2018). Moreover, we empirically validate our intuition: features from in-distribution images are in average less predictive from each other compared to pairs of features from OOD images. $w$ can perform alone as a OOD-detector, but is best used in complement to DICE. In DICE×$w$, logits are multiplied by the sigmoid output of $w$ averaged over all pairs. Table 7 shows that temperature scaling improves all approaches without modifying ranking. Finally, DICE×$w$, even without TS, is better than DICE, even with TS.

Table 6: **Out-of-distribution** performances **before** temperature scaling.

| Train | Test OOD | FPR (95 % TPR) ↓ | | | | | Detection ↓ | | | | | AUROC ↑ | | | | | AUPR In ↑ | | | | | AUPR Out ↑ | | | | |
|---|---|---|---|---|---|---|---|---|---|---|---|---|---|---|---|---|---|---|---|---|---|---|---|---|---|---|
| | | Ind | CEB | DICE | w only | DICE×w | Ind | CEB | DICE | w only | DICE×w | Ind | CEB | DICE | w only | DICE×w | Ind | CEB | DICE | w only | DICE×w | Ind | CEB | DICE | w only | DICE×w |
| CIFAR-100 | TinyImageNet (crop) | 80.1 | 80.4 | 77.9 | 82.2 | **73.7** | 33.0 | 32.1 | 31.2 | 32.4 | **28.8** | 72.4 | 73.8 | 74.7 | 71.1 | **77.2** | 71.5 | 73.0 | 73.4 | 66.0 | **74.3** | 70.5 | 71.7 | 72.4 | 66.4 | **75.9** |
| | TinyImageNet (resize) | 84.4 | 83.6 | 81.0 | 87.9 | **78.8** | 35.5 | 34.5 | 33.6 | 35.7 | **31.7** | 69.1 | 70.6 | 71.7 | 66.1 | **73.6** | 68.0 | 69.3 | **70.4** | 60.5 | 70.3 | 66.8 | 68.5 | 69.3 | 64.4 | **71.9** |
| | LSUN (crop) | 79.1 | 82.7 | 81.1 | 74.9 | **73.3** | 28.6 | 29.2 | 29.3 | 29.9 | **28.5** | 77.7 | 76.2 | 75.9 | 76.3 | **78.9** | **79.9** | 78.6 | 77.8 | 74.7 | 79.2 | 73.9 | 71.8 | 71.8 | 75.1 | **76.8** |
| | LSUN (resize) | 83.1 | 81.0 | 80.0 | 82.5 | **75.9** | 34.2 | 32.1 | 31.4 | 32.0 | **29.2** | 71.5 | 74.2 | 74.2 | 71.6 | **77.1** | 73.2 | 75.5 | 75.4 | 66.3 | **76.5** | 68.3 | 71.5 | 71.2 | 69.7 | **75.0** |
| | iSUN | 85.3 | 83.4 | 83.8 | 84.2 | **79.7** | 35.3 | 33.2 | 33.3 | 34.3 | **31.7** | 69.6 | 72.5 | 71.9 | 68.7 | **74.4** | 72.7 | **75.3** | 74.7 | 65.6 | 74.7 | 63.7 | 66.9 | 65.7 | 64.9 | **69.5** |
| | TinyImageNet+LSUN+iSUN | 82.3 | 82.2 | 80.7 | 82.3 | **76.2** | 33.4 | 32.2 | 31.8 | 32.8 | **30.2** | 72.1 | 73.5 | 73.8 | 70.9 | **76.4** | 38.1 | 39.9 | **40.3** | 28.7 | 39.4 | 91.5 | 91.9 | 91.9 | 91.4 | **93.1** |
| | CIFAR-10 | 80.1 | 82.9 | **78.5** | 90.0 | 79.9 | 30.0 | 30.3 | 28.8 | 36.6 | **28.7** | 76.6 | 76.0 | 78.1 | 66.5 | **78.4** | 79.7 | 79.1 | **80.9** | 64.6 | 80.8 | 72.5 | 71.6 | **73.9** | 62.8 | 73.9 |

Table 7: **Out-of-distribution** performances **after** temperature scaling.

| Train | Test OOD | FPR (95 % TPR) ↓ | | | | Detection ↓ | | | | AUROC ↑ | | | | AUPR In ↑ | | | | AUPR Out ↑ | | | |
|---|---|---|---|---|---|---|---|---|---|---|---|---|---|---|---|---|---|---|---|---|---|
| | | Ind | CEB | DICE | DICE×w | Ind | CEB | DICE | DICE×w | Ind | CEB | DICE | DICE×w | Ind | CEB | DICE | DICE×w | Ind | CEB | DICE | DICE×w |
| CIFAR-100 | TinyImageNet (crop) | 77.7 | 78.4 | 77.1 | **73.2** | 32.1 | 31.2 | 31.5 | **27.9** | 74.0 | 74.8 | 75.2 | **77.9** | 72.6 | 74.1 | 74.0 | **74.8** | 72.3 | 73.4 | 73.4 | **76.4** |
| | TinyImageNet (resize) | 83.0 | 82.3 | 80.4 | **78.5** | 34.4 | 33.6 | 32.9 | **30.7** | 70.5 | 71.5 | 72.6 | **74.3** | 69.0 | 70.5 | **70.9** | 70.8 | 68.3 | 70.2 | 70.3 | **72.5** |
| | LSUN (crop) | 76.7 | 81.7 | 80.2 | **72.7** | 27.6 | 28.6 | 28.5 | **28.2** | 79.3 | 77.2 | 77.0 | **79.2** | **81.2** | 79.4 | 78.7 | 79.5 | 75.9 | 73.1 | 72.5 | **77.1** |
| | LSUN (resize) | 81.5 | 79.0 | 79.5 | **75.4** | 33.1 | 30.9 | 30.6 | **28.3** | 73.0 | 75.4 | 75.1 | **78.1** | 74.3 | 76.8 | 76.0 | **76.9** | 70.1 | 73.4 | 72.2 | **75.6** |
| | iSUN | 84.3 | 82.3 | 83.2 | **79.3** | 34.5 | 32.3 | 32.2 | **30.8** | 70.9 | 73.5 | 72.7 | **75.0** | 73.5 | **76.4** | 74.6 | 75.6 | 65.2 | 68.6 | 66.6 | **70.1** |
| | TinyImageNet+LSUN+iSUN | 80.5 | 80.7 | 80.0 | **75.8** | 32.3 | 31.3 | 31.2 | **29.3** | 73.6 | 74.4 | 74.7 | **76.9** | 39.3 | **41.4** | 40.9 | 39.8 | 92.0 | 92.5 | 92.2 | **93.3** |
| | CIFAR-10 | 78.8 | 82.1 | **78.2** | 80.0 | 29.6 | 29.9 | 28.6 | **28.3** | 77.5 | 76.7 | 78.5 | **78.6** | 80.2 | 79.4 | **81.2** | 81.0 | 73.5 | 72.5 | **74.5** | 74.0 |

## A.3 ACCURACY VERSUS SIZE

We recover from Table 8 the Memory Split Advantage (MSA) from Chirkova et al. (2020): splitting the memory budget between three branches of ResNet-32 results in better performance than spending twice the budget on one ResNet-110. DICE further improves this advantage. Our framework is particularly effective in the branch-based setting, as it reduces the computational overhead (especially in terms of FLOPS) at a slight cost in diversity. A 4-branches DICE ensemble has the same accuracy in average as a classical 7-branches ensemble.

Table 8: **Ensemble effectiveness evaluation**. Top-1 accuracy (%), number of parameters (M) and floating point operations (GFLOPs). This table is summarized in Figure 1. DICE always outperforms the independent learning baseline, even with only 1 member because of the CEB component. The saturation phenomenon is reduced.

| Architecture | | | CIFAR-100 | | | |
|---|---|---|---|---|---|---|
| Backbone | Structure | Ens. Size | Params. (M) | GFLOPs | Ind. | DICE |
| | Base | 1 | 0.47 | 0.14 | 71.28 | **71.31** |
| | | 2 | 0.83 | 0.18 | 74.89 | **75.40** |
| | | 3 | 1.19 | 0.23 | 76.28 | **76.89** |
| | | 4 | 1.55 | 0.28 | 76.78 | **77.51** |
| | Branches | 5 | 1.91 | 0.32 | 77.24 | **78.08** |
| | | 6 | 2.27 | 0.36 | 77.39 | **78.29** |
| | | 7 | 2.63 | 0.40 | 77.52 | **78.44** |
| | | 8 | 2.99 | 0.44 | 77.60 | **78.60** |
| ResNet-32 | | 10 | 3.71 | 0.51 | 77.64 | **78.71** |
| | | 2 | 0.95 | 0.28 | 75.01 | **75.32** |
| | | 3 | 1.42 | 0.42 | 76.45 | **76.98** |
| | | 4 | 1.89 | 0.56 | 77.38 | **77.92** |
| | Nets | 5 | 2.36 | 0.70 | 77.82 | **78.41** |
| | | 6 | 2.83 | 0.84 | 78.16 | **78.83** |
| | | 7 | 3.29 | 0.98 | 78.36 | **79.05** |
| | | 8 | 3.78 | 1.12 | 78.49 | **79.24** |
| | | 10 | 4.71 | 1.41 | 78.59 | **79.35** |
| | Base | 1 | 1.73 | 0.51 | 76.21 | **76.25** |
| ResNet-110 | Branches | 3 | 4.33 | 0.84 | 80.54 | **81.67** |
| | | 4 | 5.68 | 1.02 | 80.89 | **81.93** |

## A.4 TRAINING DYNAMICS IN TERMS OF ACCURACY, UNCERTAINTY ESTIMATION AND CALIBRATION

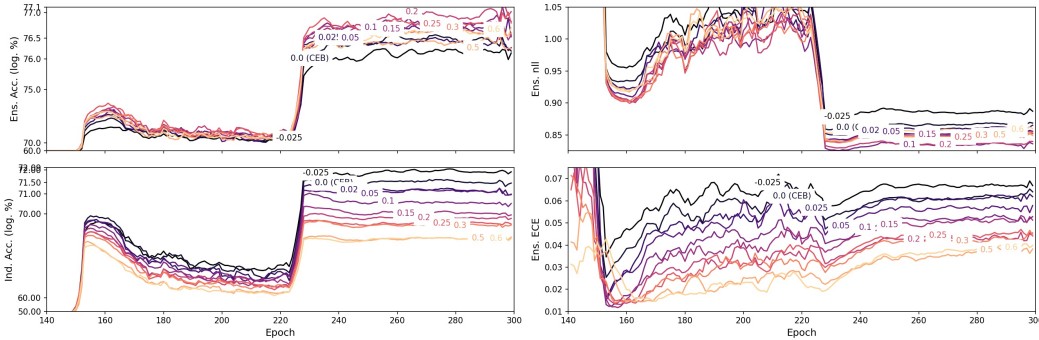

Figure 8: **Training dynamics on the validation dataset** while training on 95% of the training dataset. A higher diversity coefficient decreases individual performance (lower left), but increases ensemble performance in terms of accuracy (upper left), uncertainty estimation (upper right) up to a value, found at $\delta_{cr} = 0.2$ for 4-branches ResNet-32. Calibration before temperature scaling (lower right) highly benefits from higher diversity. Learning rate updates create "steps" in the curves.

## A.5 TRAINING DYNAMICS IN TERMS OF DIVERSITY

Figure 9: **Diversity dynamics** on train and validation dataset. DICE increases diversity for pairwise (ratio errors, agreement, Q-statistics) and non-pairwise (entropy, Kohavi-Wolpert variance) measures.

We measured diversity in 4.b with the ratio error (Aksela, 2003). But as stated by Kuncheva & Whitaker (2003), diversity can be measured in numerous ways. For pairwise measures, we averaged over the $\binom{M}{2}$ pairs: the Q-statistics is positive when classifiers recognize the same object, the agreement score measures the frequency that both classifiers predict the same class. Note that even if we only apply pairwise constraints, we also increase non-pairwise measures: for example, the Kohavi-Wolpert variance (Kohavi et al., 1996) which measures the variability of the predicted class, and the entropy diversity which measures overall disagreement.

## A.6 UNCERTAINTY ESTIMATION AND CALIBRATION BEFORE TEMPERATURE SCALING

Table 9: **Uncertainty estimation** (NLL, BS) and **calibration** (ECE, TACE) on CIFAR-100 **before** temperature scaling for 4-branches ResNet-32.

| Metrics | 1-net | Ind. | OKDDip-E | ADP | IB | CEB | IBR | CEBR | DICE |
|---|---|---|---|---|---|---|---|---|---|
| NLL $\downarrow (10^{-1})$ | 11.56 | 8.55 | 8.38 | 10.85 | 8.37 | 8.37 | 8.27 | 8.25 | **8.06** |
| BS $\downarrow (10^{-3})$ | 4.12 | 3.35 | 3.28 | 3.79 | 3.25 | 3.25 | 3.21 | 3.23 | **3.15** |
| ECE $\downarrow (10^{-2})$ | 10.47 | 7.45 | 6.67 | 21.14 | 5.32 | 5.76 | 5.15 | 5.46 | **4.05** |
| TACE $\downarrow (10^{-3})$ | 2.42 | 1.86 | 1.81 | 4.53 | 1.58 | 1.67 | 1.60 | 1.65 | **1.46** |
| Acc. $\uparrow (\%)$ | 71.28 | 76.71 | 76.85 | 77.21 | 76.93 | 76.98 | 77.25 | 77.30 | **77.51** |

## B   TRAINING DETAILS

### B.1   GENERAL OPTIMIZATION

**Experiments**   Classical hyperparameters were taken from (Chen et al., 2020b) for conducting fair comparisons. Newly added hyperparameters were fine-tuned on a validation dataset made of 5% of the training dataset.

**Architecture**   We implemented the proposed method with ResNet (He et al., 2016) and Wide-ResNet (Zagoruyko & Komodakis, 2016) architectures. Following standard practices, we average the logits of our predictions uniformly. For branch-based ensemble, we separate the last block and the classifier of each member from the weights sharing while the other low-level layers were shared.

**Learning**   Following (Chen et al., 2020b), we used SGD with Nesterov with momentum of 0.9, mini-batch size of 128, weight decay of 5e-4, 300 epochs, a standard learning rate scheduler that sets values $\{0.1, 0.001, 0.0001\}$ at steps $\{0, 150, 225\}$ for CIFAR-10/100. In CIFAR-100, we additionally set the learning rate at 0.00001 at step 250. We used traditional basic data augmentation that consists of horizontal flips and a random crop of 32 pixels with a padding of 4 pixels. The learning curve is shown on Figure 8.

### B.2   INFORMATION BOTTLENECK IMPLEMENTATION

**Architecture**   Features are extracted just before the dense layer since deeper layers are more semantics, of size $d = \{64, 128, 256\}$ for {ResNet-32, WRN-28-2, ResNet-110}. Our encoder does not provide a deterministic point in the features space but a feature distribution encoded by mean and diagonal covariance matrix. The covariance is predicted after a Softplus activation function with one additional dense layer, taking as input the features mean, with $d(d + 1)$ trainable weights. In training we sample once from this features distribution with the reparameterization trick. In inference, we predict from the distribution's mean (and therefore only once). We parameterized $b(z|y) \sim N(b^\mu(y), \mathbf{1})$ with trainable mean and unit diagonal covariance, with $d$ additional trainable weights per class. As noticed in (Fischer & Alemi, 2020), this can be represented as a single embedding layer mapping one-hot classes to $d$-dimensional tensors. Therefore in total we only add $d(d + 1 + K)$ trainable weights, that all can be discarded during inference. For VIB, the embedding $b^\mu$ is shared among classes: in total it adds $d(d + 2)$ trainable weights. Contrary to recent IB approaches (Wu et al., 2019b; Wu & Fischer, 2020; Fischer & Alemi, 2020), we only have one dense layer to predict logits after the features bottleneck, and we did not change the batch normalization, for fair comparisons with traditional ensemble methods.

**Scheduling**   We employ the jump-start method that facilitates the learning of bottleneck-inspired models (Wu et al., 2019b; Wu & Fischer, 2020; Fischer & Alemi, 2020): we progressively anneal the value of $\beta_{ceb}$. For CIFAR-10, we took the scheduling from (Fischer & Alemi, 2020), except that we widened the intervals to make the training loss decrease more smoothly: $\log(\beta_{ceb})$ reaches values $\{100, 10, 2\}$ at steps $\{0, 5, 100\}$. No standard scheduling was available for CIFAR-100. As it is more difficult than CIFAR-10, we added additional jump-epochs with lower values: $\log(\beta_{ceb})$ reaches values $\{100, 10, 2, 1.5, 1\}$ at steps $\{0, 8, 175, 250, 300\}$. This slow scheduling increases progressively the covariance predictions $e^\sigma(x)$ and facilitates learning. For VIB, we scheduled similarly using the equivalence from (Fischer, 2020): $\beta_{ib} = \beta_{ceb} + 1$. We found VCEB to have lower standard deviation in performances than VCEB: $\beta_{ib}$ can hinder the learnability (Wu et al., 2019b). These schedulings have been used in all our setups, without and with redundancy losses, for ResNet-32, ResNet-110 and WRN-28-10, for from 1 to 10 members.

### B.3   ADVERSARIAL TRAINING IMPLEMENTATION

**Redundancy**   Following standard adversarial learning practices, our discriminator for redundancy estimation is a MLP with 4 layers of size $\{256, 256, 100, 1\}$, with leaky-ReLus of slope 0.2, optimized by RMSProp with learning rate $\{0.003, 0.005\}$ for CIFAR-$\{10, 100\}$. We empirically found that four steps for the discriminator for one step of the classifier increase stability. Specifically, it takes as input the concatenation of the two hidden representations of size $d$, sampled with a repa-

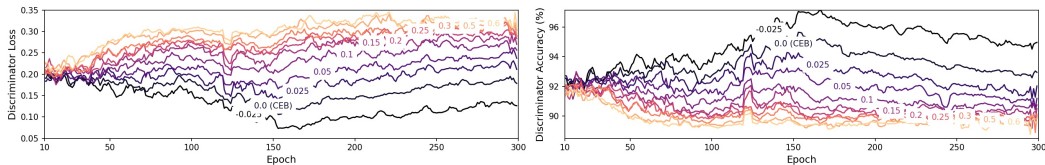

Figure 10: **Discriminator** dynamics and learning curve. The task becomes harder for higher values of $\delta_{cr}$: the joint and product features distributions tend to be indistinguishable.

rameterization trick. Gradients are not backpropagated in the layer that predicts the covariance, as it would artificially increase the covariance to reduce the mutual information among branches. The output, followed by a sigmoid activation function, should be close to 1 (resp. 0) if the sample comes from the joint (resp. product) distribution.

**Conditional Redundancy** The discriminator for CR estimation needs to take into account the target class $Y$. It first embeds $Y$ in an embedding layer of size 64, which is concatenated at the inputs of the first and second layers. Improved features merging method could be applied, such as Ben-Younes et al. (2019). The output has size $K$, and we select the index associated with the $Y$. We note in Figure 11 that our discriminator remains calibrated.

Figure 11: **The discriminator remains calibrated** even at the end of the adversarial training.

**Ensemble with $M$ Models** In the general case, we only consider pairwise interactions, therefore we need to estimate $\binom{M}{2}$ values. To reduce the number of parameters, we use only one discriminator $w$. Features associated with $z_k$ are filled with zeros when we sample from $p(z_i, z_j, y)$ or from $p(z_i, y)p(z_j|y)$, where $i, j, k \in \{1, \dots, M\}, k \neq i$ and $k \neq j$. Therefore, the input tensor for the discriminator is of size $(M * d + 64)$: its first layer has $(M * d + 64) * 256$ dense weights: the number of weights in $w$ scales linearly with $M$ and $d$ as $w$'s input grows linearly, but $w$'s hidden size remains fixed.

$\delta_{cr}$ **value** For branch-based and network-based CIFAR-100, we found $\delta_{cr}$ at $\{0.1, 0.15, 0.2, 0.22, 0.25\}$ for $\{2, 3, 4, 5, 6\}$ members to perform best on the validation dataset when training on 95% on the classical training dataset. For CIFAR-10, $\{0.1\}$ for 4 members. We found that lower values of $\delta_r$ were necessary for our baselines IBR and CEBR.

**Scheduling** For fair comparison, we apply the traditional ramp-up scheduling up to step 80 from the co-distillation literature (Lan et al., 2018; Kim et al., 2019b; Chen et al., 2020b) to all concurrent approaches and to our redundancy training.

**Sampling** To sample from $p(z_1, z_2, y)$, we select features extracted from one image. To sample from $p(z_1, y)p(z_2|y)$, we select features extracted from two different inputs, that share the same class $y$. In practise, we keep a memory from previous batches as the batch size is 128 whereas we have 100 classes in CIFAR-100. This memory, of size $M * d * K * 4$, is updated at the end of each training step. Our sampling is a special case of $k$-NN sampling (Molavipour et al., 2020): as we sample from a discrete categorical variable, the closest neighbour has exactly the same discrete value. The training can be unstable as it minimises the divergence between two distributions. To make them overlap over the features space, we sample $num_{sample} = \{4\}$ times from the gaussian distribution of $Z_1$ and $Z_2$ with the reparameterization trick. This procedure is similar to instance noise (Sønderby et al., 2016) and it allows us to safely optimise $w$ at each iteration. It gives better robustness than just giving the gaussian mean. Moreover, we progressively ease the discriminator task by scheduling the covariance through time with a linear ramp-up. First the covariance is set to $\mathbf{1}$ until epoch 100, then it linearly reduces to the predicted covariance $e_i^\sigma(x)$ until step 250. We sample a ratio $ratio_{pos}^{neg}$ of one positive pair for $\{2, 4\}$ negative pairs on CIFAR-$\{10, 100\}$.

**Clipping** Following Bachman et al. (2019), we clip the density ratios ($tanhclip$) by computing the non linearity $\exp[\tau \tanh \frac{\log[f(z_1,z_2,y)]}{\tau}]$. A lower $\tau$ reduces the variance of the estimation and stabilizes the training even with a strong discriminator, at the cost of additional bias. The clipping threshold $\tau$ was set to 10 as in Song & Ermon (2020).

## B.4 Pseudo-Code

---

**Algorithm 1:** Full DICE Procedure for $M = 2$ members

---

```
/* Setup                                                              */
```
**Parameters:** $\theta_1 = \{e_1, b_1, c_1\}$, $\theta_2 = \{e_2, b_2, c_2\}$ and discriminator $w$, randomly initialized
**Input:** Observations $\{x^n, y^n\}_{n=1}^N$, coefficients $\beta_{ceb}$ and $\delta_{cr}$, schedulings $sche_{ceb}$ and
$\quad\quad rampup_{startstep}^{endstep}$, clipping threshold $\tau$, batch size $b$, optimisers $g_{\theta_{1,2}}$ and $g_w$,
$\quad\quad$ number of discriminators step $nstep_d$, number of samples $num_s$, ratio of
$\quad\quad$ positive/negative sample $ratio_{pos}^{neg}$

```
/* Training Procedure                                                 */
```
**1** **for** $s \leftarrow 1$ **to** $300$ **do**
**2** $\quad$ $\beta_{ceb}^s \leftarrow sche_{ceb}(\text{startvalue=0, endvalue=}\beta_{ceb}, \text{step=}s)$
**3** $\quad$ $\delta_{cr}^s \leftarrow rampup_0^{80}(\text{startvalue=0, endvalue=}\delta_{cr}, \text{step=}s)$
**4** $\quad$ Randomly select batch $\{(x^n, y^n)\}_{n\in\mathcal{B}}$ of size $b$ $\quad\quad\quad$ `// Batch Sampling`
```
   /* Step 1:  Classification Loss with CEB                          */
```
**5** $\quad$ **for** $m \leftarrow 1$ **to** $2$ **do**
**6** $\quad\quad$ $z_i^n \leftarrow e_i^\mu(z|x^n) + \epsilon e_i^\sigma(z|x^n), \forall n \in \mathcal{B}$ with $\epsilon \sim N(0,1)$
**7** $\quad\quad$ $\text{VCEB}_i \leftarrow \frac{1}{b} \sum_{n\in\mathcal{B}} \{\frac{1}{\beta_{ceb}^s} D_{\text{KL}}(e_i(z|x^n) \| b_i(z|y^n)) - \log c_i(y^n|z_i^n\}$

```
   /* Step 2:  Diversity Loss with Conditional Redundancy            */
```
**8** $\quad$ **for** $m \leftarrow 1$ **to** $2$ **do**
**9** $\quad\quad$ $e_i^{\sigma,s}(z|x^n) = rampup_{100}^{250}(\text{startvalue=1, endvalue=}e_i^\sigma(z|x^n), \text{step=}s)$
**10** $\quad\quad$ **for** $k \leftarrow 1$ **to** $num_s$ **do**
**11** $\quad\quad\quad$ $z_{i,k}^n \leftarrow e_i^\mu(z|x^n) + \epsilon e_i^{\sigma,s}(z|x^n), \forall n \in \mathcal{B}$ with $\epsilon \sim N(0,1)$

**12** $\quad$ $\mathcal{B}_\text{J} \leftarrow \{(z_{1,k}^n, z_{2,k}^n, y^n)\}, \forall n \in \mathcal{B}, k \in \{1, \ldots, num_s\}$ $\quad\quad$ `// Joint Distrib.`
**13** $\quad$ $\hat{\mathcal{L}}_{DV}^{CR} \leftarrow \frac{1}{|\mathcal{B}_\text{J}|} \sum_{t\in\mathcal{B}_\text{J}} \log f(t)$ with $f(t) \leftarrow \text{tanhclip}(\frac{w(t)}{1-w(t)}, \tau)$

**14** $\quad$ $\theta_{1,2} \leftarrow g_{\theta_{1,2}}(\nabla_{\theta_1}\text{VCEB}_1 + \nabla_{\theta_2}\text{VCEB}_2 + \delta_{cr}^s \nabla_{\theta_{1,2}} \hat{\mathcal{L}}_{DV}^{CR})$ $\quad$ `// Backprop Ensemble`
```
   /* Step 3:  Adversarial Training                                  */
```
**15** $\quad$ **for** $\_ \leftarrow 1$ **to** $nstep_d$ **do**
**16** $\quad\quad$ $\mathcal{B}_\text{J} \leftarrow \{(z_{1,k}^n, z_{2,k}^n, y^n)\}, \forall n \in \mathcal{B}, \forall k \in \{1, \ldots, num_s\}$ $\quad\quad$ `// Joint Distrib.`
**17** $\quad\quad$ $\mathcal{B}_p \leftarrow \{(z_{1,k}^n, z_{2,k'}^{n'}, y^n)\}, \forall n \in \mathcal{B}, \forall k \in \{1, \ldots, num_s\}, k' \in \{1, \ldots, ratio_{pos}^{neg}\}$
**18** $\quad\quad$ with $n' \in \mathcal{B}, y^n = y^{n'}, n \neq n'$ $\quad\quad\quad\quad$ `// Product distribution`
**19** $\quad\quad$ $w \leftarrow g_w(\nabla_w \mathcal{L}_{ce}(w))$ $\quad\quad\quad\quad$ `// Backprop Discriminator`
**20** $\quad\quad$ Sample new $z_{i,k}^n$

```
/* Test Procedure                                                     */
```
**Data:** Inputs $\{x^n\}_{n=1}^T$ $\quad\quad\quad\quad$ `// Test Data`
**Output:** $\arg\max_{k\in\{1,\ldots,K\}} (\frac{1}{2}[c_1(e_1^\mu(z|x^n)) + c_2(e_2^\mu(z|x^n))]), \forall n \in \{1, \ldots, T\}$

---

## B.5 Empirical Limitations

Our approach relies on very recent works in neural network estimation of mutual information, that still suffer from loose approximations. Improvements in this area would facilitate our learning procedure. Our approach increases the number of operations because of the adversarial procedure, but only during training: the inference time remains the same.

Table 10: **Summary of different approaches**.

| Method | Co-distillation | Diversity | I.B. | Merging | Others | Branch/Net | Consistently better than Ind. |
|---|---|---|---|---|---|---|---|
| DML (Zhang et al., 2018) | Pred. pairwise | | | | | Net | |
| CL-ILR (Song & Chai, 2018) | Pred. | | | | | Branch | |
| ONE (Lan et al., 2018) | Preds | | | Gate | | Branch | |
| FFL (Kim et al., 2019b) | Pred. | | | Feat. Fus. | | Both | |
| OKDDip (Chen et al., 2020b) | Pred. asymetric | | | | | Both | ≈ |
| KDCL (Guo et al., 2020) | Pred. | Data Augmentation | | Weights on val | | Net | |
| PCL (Wu & Gong, 2020) | Pred. | Data Augmentation | | Feat. Fus. | Mean teacher | Branch | ≈ |
| AFD (Chung et al., 2020) | Features | | | | | Net | ≈ |
| GAL (Kariyappa & Qureshi, 2019) | | Gradients | | | | Net | |
| GPMR (Dabouei et al., 2020) | | Gradients | | | Grads. Magnitude | Net | |
| ADP (Pang et al., 2019) | | Non maximum pred. | | | Entropy Pred. | Both | ✓ |
| DIBS (Sinha et al., 2020) | | JSD Features | VIB | | Custom sampling | Both | ? |
| IB | | | VIB | | | Both | ✓ |
| CEB | | | VCEB | | | Both | ✓ |
| IBR (Ours equation 9) | R | | VIB | | | Both | ✓ |
| CEBR (Ours equation 10) | R | | VCEB | | | Both | ✓ |
| DICE (Ours equation 5) | CR | | VCEB | | | Both | ✓ |

# C    CONCURRENT APPROACHES

Concurrent approaches can be divided in two general patterns: they promote either individual accuracy by co-distillation either ensemble diversity.

## C.1    CO-DISTILLATION APPROACHES

Contrary to the traditional distillation (Hinton et al., 2015) that aligns the soft prediction between a static pre-trained strong teacher towards a smaller student, online co-distillation performs teaching in an end-to-end one-stage procedure: the teacher and the student are trained simultaneously.

**Distillation in Logits**    The seminal "Deep Mutual Learning" (DML) (Zhang et al., 2018) introduced the main idea: multiple networks learn to mimic each other by reducing KL-losses between pairs of predictions. "Collaborative learning for deep neural networks" (CL-ILR) (Song & Chai, 2018) used the branch-based architecture by sharing low-level layers to reduce the training complexity, and "Knowledge Distillation by On-the-Fly Native Ensemble" (ONE) (Lan et al., 2018) used a weighted combination of logits as teacher hence providing better information to each network. "Online Knowledge Distillation via Collaborative Learning" (KDCL) (Guo et al., 2020) computed the optimum weight on an held-out validation dataset. "Feature Fusion for Online Mutual Knowledge Distillation" (FFL) (Kim et al., 2019b) introduced a feature fusion module. These approaches improve individual performance at the cost of increased homogenization. "Online Knowledge Distillation with Diverse Peers" (OKDDip) (Chen et al., 2020b) slightly alleviates this problem with an asymmetric distillation and a self-attention mechanism. "Peer Collaborative Learning for Online Knowledge Distillation" (PCL) (Wu & Gong, 2020) benefited from the mean-teacher paradigm with temporal ensembling and from diverse data augmentation, at the cost of multiple inferences through the shared backbone.

**Distillation in Features**    Whereas all previous approaches only apply distillation on the logits, the recent "Feature-map-level Online Adversarial Knowledge Distillation" (AFD) (Chung et al., 2020) aligned features distributions by adversarial training. Note that this is not opposite to our approach, as they force distributions to be similar while we force them to be uncorrelated.

## C.2    DIVERSITY APPROACHES

On the other hands, some recent papers in computer vision explicitly encourage diversity among the members with regularization losses.

**Diversity in Logits**    "Diversity Regularization in Deep Ensembles" (Shui et al., 2018) applied negative correlation (Liu & Yao, 1999a) to regularize the training for improved calibration, with no impact on accuracy. "Learning under Model Misspecification: Applications to Variational and Ensemble methods" (Masegosa, 2020) theoretically motivated the minimization of second-order PAC-Bayes bounds for ensembles, empirically estimated through a generalized variational method.

"Adaptive Diversity Promoting" (ADP) (Pang et al., 2019) decorrelates only the non-maximal predictions to maintain the individual accuracies, while promoting ensemble entropy. It forces different members to have different ranking of predictions among non maximal predictions. However, Liang et al. (2018) has shown that ranking of outputs are critical: for example, non maximal logits tend to be more separated from each other for in-domain inputs compared to out-of-domain inputs. Therefore individual accuracies are decreased. Coefficients $\alpha$ and $\beta$ are respectively set to 2 and 0.5, as in the original paper.

**Diversity in Features**  One could think about increasing classical distances among features like $L_2$ in (Kim et al., 2018), but in our experiments it reduces overall accuracy: it is not even invariant to linear transformations such as translation. "Diversity inducing Information Bottleneck in Model Ensembles" from Sinha et al. (2020) trains a multi-branch network and applies VIB on individual branch, by encoding $p(z|y) \sim \mathcal{N}(0, 1)$, which was shown to be hard to learn (Wu & Fischer, 2020). Moreover, we notice that their diversity-inducing adversarial loss is an estimation of the JS-divergence between pairs of features, built on the dual $f$-divergence representation (Nowozin et al., 2016): similar idea was recently used for saliency detection (Chen et al., 2020a). As the JS-divergence is a symmetrical formulation of the KL, we argue that DIBS and IBR share the same motivations and only have minor discrepancies: the adversarial terms in DIBS loss with both terms sampled from the same branch and both terms sampled from the same prior. In our experiments, these differences reduce overall performance. We will include their scores when they publish measurable results on CIFAR datasets or when they release their code.

**Diversity in Gradients**  "Improving adversarial robustness of ensembles with diversity training." (GAL) (Kariyappa & Qureshi, 2019) enforced diversity in the gradients with a gradient alignment loss. "Exploiting Joint Robustness to Adversarial Perturbations" (Dabouei et al., 2020) considered the optimal bound for the similarity of gradients. However, as stated in the latter, "promoting diversity of gradient directions slightly degrades the classification performance on natural examples . . . [because] classifiers learn to discriminate input samples based on distinct sets of representative features". Therefore we do not consider them as concurrent work.

# D  EXPERIMENTAL SETUP

## D.1  TRAINING DATASETS

We train our procedure on two image classification benchmarks, CIFAR-100 and CIFAR-10, (Krizhevsky et al., 2009). They consist of 60k 32*32 natural and colored images in respectively 100 classes and 10 classes, with 50k training images and 10k test images. For hyperparameter selection and ablation studies, we train on 95% of the training dataset, and analyze performances on the validation dataset made of the remaining 5%.

## D.2  OOD

**Dataset**  We used the traditional out-of-distribution datasets for CIFAR-100, described in (Liang et al., 2018): TinyImageNet (Deng et al., 2009), LSUN (Yu et al., 2015), iSUN(Xu et al., 2015), and CIFAR-10. We borrowed the evaluation code from `https://github.com/uoguelph-mlrg/confidence_estimation` (DeVries & Taylor, 2018).

**Metrics**  We reported the standard metrics for binary classification: FPR at 95 % TPR, Detection error, AUROC (Area Under the Receiver Operating Characteristic curve) and AUPR (Area under the Precision-Recall curve, -in or -out depending on which dataset is specified as positive). See Liang et al. (2018) for definitions and interpretations of these metrics.

# E ADDITIONAL THEORETICAL ELEMENTS

## E.1 BIAS VARIANCE COVARIANCE DECOMPOSITION

The Bias-Variance-Covariance Decomposition (Ueda & Nakano, 1996) generalizes the Bias-Variance Decomposition (Kohavi et al., 1996) by treating the ensemble of M members as a single learning unit.

$$\mathbb{E}[(\overline{f} - t)^2] = \overline{\text{bias}}^2 + \frac{1}{M}\overline{\text{var}} + (1 - \frac{1}{M})\overline{\text{covar}}, \tag{7}$$

with

$$\overline{\text{bias}} = \frac{1}{M}\sum_i (\mathbb{E}[f_i] - t),$$

$$\overline{\text{var}} = \frac{1}{M}\sum_i \mathbb{E}[(\mathbb{E}[f_i] - t)^2],$$

$$\overline{\text{covar}} = \frac{1}{M(M-1)}\sum_i \sum_{j \neq i} \mathbb{E}[(f_i - \mathbb{E}[f_i])(f_j - \mathbb{E}[f_j])].$$

The estimation improves when the covariance between members is zero: the reduction factor of the variance component equals to M when errors are uncorrelated. Compared to the Bias-Variance Decomposition (Kohavi et al., 1996), it leads to a variance reduction of $\frac{1}{M}$. Brown et al. (2005a;b) summarized it this way: "in addition to the bias and variance of the individual estimators, the generalisation error of an ensemble also depends on the covariance between the individuals. This raises the interesting issue of why we should ever train ensemble members separately; why shouldn't we try to find some way to capture the effect of the covariance in the error function?".

## E.2 MUTUAL INFORMATION

> Nobody knows what entropy really is.
>
> *John Van Neumann* to *Claude Shannon*

At the cornerstone of Shannon's information theory in 1948 (Shannon, 1948), mutual information is the difference between the sum of individual entropies and the entropy of the variables considered jointly. Stated otherwise, it is the reduction in the uncertainty of one variable due to the knowledge of the other variable (Cover, 1999). Entropy owed its name to the thermodynamic measure of uncertainty introduced by Rudolf Clausius and developed by Ludwig Boltzmann.

$$I(Z_1; Z_2) = H(Z_1) + H(Z_2) - H(Z_1, Z_2)$$
$$= H(Z_1) - H(Z_1|Z_2)$$
$$= D_{\text{KL}}(P(Z_1, Z_2)\|P(Z_1)P(Z_2)).$$

The conditional mutual information generalizes mutual information when a third variable is given:

$$I(Z_1; Z_2|Y) = D_{\text{KL}}(P(Z_1, Z_2|Y)\|P(Z_1|Y)P(Z_2|Y)).$$

## E.3 KL BETWEEN GAUSSIANS

The Kullback-Leibler divergence (Kullback, 1959) between two gaussian distributions takes a particularly simple form:

$$D_{\text{KL}}(e(z|x)\|b(z|y)) = \log\frac{b^\sigma(y)}{e^\sigma(x)} + \frac{e^\sigma(x)^2 + (e^\mu(x) - b^\mu(y))^2}{2b^\sigma(y)^2} - \frac{1}{2} \quad \text{(Gaussian param.)}$$

$$= \frac{1}{2}[\underbrace{(1 + e^\sigma(x)^2 - \log(e^\sigma(x)^2))}_{\text{Variance}} + \underbrace{(e^\mu(x) - b^\mu(y))^2}_{\text{Mean}}]. \quad (b^\sigma(y) = \mathbf{1})$$

The variance component forces the predicted variance $e^\sigma(x)$ to be close to $b^\sigma(y) = \mathbf{1}$. The mean component forces the class-embedding $b^\mu(y)$ to converge to the average of the different elements

in its class. These class-embeddings are similar to class-prototypes, highlighting a theoretical link between CEB (Fischer, 2020; Fischer & Alemi, 2020) and prototype based learning methods (Liu & Nakagawa, 2001).

### E.4 DIFFERENCE BETWEEN VCEB AND VIB

In Fischer (2020), CEB is variationally upper bounded by VCEB. We detail the computations:

$$
\begin{aligned}
\text{CEB}_{\beta_{ceb}}(Z) &= \frac{1}{\beta_{ceb}} I(X; Z|Y) - I(Y; Z) && \text{(Definition)} \\
&= \frac{1}{\beta_{ceb}} [I(X, Y; Z) - I(Y; Z)] - I(Y; Z) && \text{(Chain rule)} \\
&= \frac{1}{\beta_{ceb}} [I(X; Z) - I(Y; Z)] - I(Y; Z) && \text{(Markov assumptions)} \\
&= \frac{1}{\beta_{ceb}} [-H(Z|X) + H(Z|Y)] - [H(Y) - H(Y|Z)] && \text{(MI as diff. of 2 ent.)} \\
&\leq \frac{1}{\beta_{ceb}} [-H(Z|X) + H(Z|Y)] - [-H(Y|Z)] && \text{(Non-negativity of ent.)} \\
&= \int \{ \frac{1}{\beta_{ceb}} \log \frac{e(z|x)}{p(z|y)} - \log p(y|z) \} p(x, y, z) \partial x \partial y \partial z && \text{(Definition of ent.)} \\
&\leq \int \{ \frac{1}{\beta_{ceb}} \log \frac{e(z|x)}{b(z|y)} - \log c(y|z) \} p(x, y) e(z|x) \partial x \partial y \partial z && \text{(Variational approx.)} \\
&\approx \frac{1}{N} \sum_{n=1}^{N} \int \{ \frac{1}{\beta_{ceb}} \log \frac{e(z|x^n)}{b(z|y^n)} - \log c(y^n|z) \} e(z|x^n) \partial z && \text{(Empirical data distrib.)} \\
&\approx \text{VCEB}_{\beta_{ceb}}(\theta = \{e, b, c\}), && \text{(Reparameterization trick)}
\end{aligned}
$$

where

$$
\text{VCEB}_{\beta_{ceb}}(\theta = \{e, b, c\}) = \frac{1}{N} \sum_{n=1}^{N} \{ \frac{1}{\beta_{ceb}} D_{\text{KL}}(e(z|x^n) \| b(z|y^n)) - \mathbb{E}_\epsilon \log c(y^n|e(x^n, \epsilon)) \}.
$$

As a reminder, Alemi et al. (2017) upper bounded: $\text{IB}_{\beta_{ib}}(Z) = \frac{1}{\beta_{ib}} I(X; Z) - I(Y; Z)$ by:

$$
\text{VIB}_{\beta_{ib}}(\theta = \{e, b, c\}) = \frac{1}{N} \sum_{n=1}^{N} \{ \frac{1}{\beta_{ib}} D_{\text{KL}}(e(z|x^n) \| b(z)) - \mathbb{E}_\epsilon \log c(y^n|e(x^n, \epsilon)) \}. \tag{8}
$$

In VIB, all features distribution $e(z|x)$ are moved towards the same class-agnostic distribution $b(z) \sim N(\mu, \sigma)$, independently of $y$. In VCEB, $e(z|x)$ are moved towards the class conditional marginal $b^\mu(y) \sim N(b^\mu(y), b^\sigma(y))$. This is the **unique difference between VIB and VCEB**. VIB leads to a looser approximation with more bias than VCEB.

### E.5 TRANSFORMING IBR AND CEBR INTO TRACTABLE LOSSES

In this section we derive the variational approximation of the IBR criterion, defined by:

$$
\text{IBR}_{\beta_{ib}, \delta_r}(Z_1, Z_2) = \text{IB}_{\beta_{ib}}(Z_1) + \text{IB}_{\beta_{ib}}(Z_2) + \delta_r I(Z_1; Z_2).
$$

**Redundancy Estimation** To estimate the redundancy component, we apply the same procedure as for conditional redundancy but without the categorical constraint, as in the seminal work of Belghazi et al. (2018) for mutual information estimation. Let $\mathcal{B}_{\text{J}}$ and $\mathcal{B}_p$ be two random batches sampled respectively from the observed joint distribution $p(z_1, z_2) = p(e_1(z|x), e_2(z|x))$ and the product distribution $p(z_1)p(z_2) = p(e_1(z|x))p(e_2(z|x'))$, where $x, x'$ are two inputs that may not belong to the same class. We similarly train a network $w$ that tries to discriminate these two distributions. With $f = \frac{w}{1-w}$, the redundancy estimation is:

$$
\hat{\mathcal{I}}_{DV}^R = \frac{1}{|\mathcal{B}_{\text{J}}|} \sum_{(z_1, z_2) \in \mathcal{B}_{\text{J}}} \underbrace{\log f(z_1, z_2)}_{\text{Diversity}} - \log(\frac{1}{|\mathcal{B}_p|} \sum_{(z_1, z_2') \in \mathcal{B}_p} f(z_1, z_2')),
$$

and the final loss:

$$
\hat{\mathcal{L}}_{DV}^R(e_1, e_2) = \frac{1}{|\mathcal{B}_{\text{J}}|} \sum_{(z_1, z_2) \in \mathcal{B}_{\text{J}}} \log f(z_1, z_2).
$$

**IBR**  Finally we train $\theta_1 = \{e_1, b_1, c_1\}$ and $\theta_2 = \{e_2, b_2, c_2\}$ jointly by minimizing:

$$\mathcal{L}_{IBR}(\theta_1, \theta_2) = \text{VIB}_{\beta_{ib}}(\theta_1) + \text{VIB}_{\beta_{ib}}(\theta_2) + \delta_r \hat{\mathcal{L}}_{DV}^R(e_1, e_2). \tag{9}$$

**CEBR**  For ablation study, we also consider a criterion that would benefit from CEB's tight approximation but with non-conditional redundancy regularization:

$$\mathcal{L}_{CEBR}(\theta_1, \theta_2) = \text{VCEB}_{\beta_{ceb}}(\theta_1) + \text{VCEB}_{\beta_{ceb}}(\theta_2) + \delta_r \hat{\mathcal{L}}_{DV}^R(e_1, e_2). \tag{10}$$

# F  FIRST, SECOND AND HIGHER-ORDER INFORMATION INTERACTIONS

## F.1  DICE REDUCES FIRST AND SECOND ORDER INTERACTIONS

Applying information-theoretic principles for deep ensembles leads to tackling interactions among features through conditional mutual information minimization. We define the order of an information interaction as the number of different extracted features involved.

**First Order**  Tackling the first-order interaction $I(X; Z_i|Y)$ with VCEB empirically increased overall performance compared to ensembles of deterministic features extractors learned with categorical cross entropy, at no cost in inference and almost no additional cost in training. In the Markov chain $Z_i \leftarrow X \rightarrow Z_j$, the chain rules provides: $I(Z_i; Z_j|Y) \leq I(X; Z_i|Y)$. More generally, $I(X; Z_i|Y)$ upper bounds higher order interactions such as third order $I(Z_i; Z_j, Z_k|Y)$. In conclusion, VCEB reduces an upper bound of higher order interactions with quite a simple variational approximation.

**Second Order**  In this paper, we directly target the second-order interaction $I(Z_i; Z_j|Y)$ through a more complex adversarial training. We increase diversity and performances by remove spurious correlations shared by $Z_i$ *and* $Z_j$ that would otherwise cause simultaneous errors.

**Higher Order**  interactions include the third order $I(Z_i; Z_j, Z_k|Y)$, the fourth order $I(Z_i; Z_j, Z_k, Z_l|Y)$, etc, up to the $M$-th order. They capture more complex correlations among features. For example, $Z_j$ alone (and $Z_k$ alone) could be unable to predict $Z_i$, while they $[Z_j, Z_k]$ could together. However we only consider first and second order interactions in the current submission. It is common practice, for example in the feature selection literature (Battiti, 1994; Fleuret, 2004; Brown, 2009; Peng et al., 2005). The main reason to truncate higher order interactions is computational, as the number of components would grow exponentially and add significant additional cost in training. Another reason is empirical, the additional hyper-parameters may be hard to calibrate. But these higher order interactions could be approximated through neural estimations like the second order. For example, for the third order, features $Z_i$, $Z_j$ and $Z_k$ could be given simultaneously to the discriminator $w$. The complete analysis of these higher order interactions has huge potential and could lead to a future research project.

## F.2  LEARNING FEATURES INDEPENDENCE WITHOUT COMPRESSION

The question is whether we could learn deterministic encoders with second order $I(Z_i; Z_j|Y)$ regularization without tackling first order $I(X; Z_i|Y)$. We summarized several approaches in Table 11.

**First Approach Without Sampling**  Deterministic encoders predict deterministic points in the features space. Feeding the discriminator $w$ with deterministic triples without sampling increases diversity and reaches 77.09, compared to 76.78 for independent deterministic. Compared to DICE, $w$'s task has been simplified: indeed, $w$ tries to separate the joint and the product deterministic distributions that may not overlap anymore. This violates convergence conditions, destabilizes overall adversarial training and the equilibrium between the encoders and the discriminator.

**Sampling and Reparameterization Trick**  To make the joint and product distributions overlap over the features space, we apply the reparametrization trick on features with variance **1**. This second approach is similar to instance noise (Sønderby et al., 2016), which tackled the instability of adversarial training. We reached 77.33 by protecting individual accuracies.

Table 11: **Comparison between deterministic and distribution encoders** on 4-branches ResNet-32 for Top-1 accuracy (%) on CIFAR-100.

| Method | CR | Reparameterization trick | Variance in Sampling | Categorical Cross-Entropy (Deterministic Encoder) | VCEB (Distribution Encoder) |
|---|---|---|---|---|---|
| No CR | | | | $76.78 \pm 0.19$ | $76.98 \pm 0.18$ |
| CR without sampling | ✓ | | | $77.09 \pm 0.24$ | $77.12 \pm 0.17$ |
| CR with variance=1 | ✓ | ✓ | **1** | $77.33 \pm 0.21$ | $77.29 \pm 0.14$ |
| CR with input-dependant variance | ✓ | ✓ | $e_i^\sigma(x)$ | - | $\mathbf{77.51} \pm 0.17$ |

**Synergy between CEB and CR** In comparison, we obtain 77.51 with DICE. In addition to theoretical motivations, VCEB and CR work empirically in synergy. *First*, the adversarial learning is simplified and only focuses on spurious correlations VCEB has not already deleted. Thus it may explain the improved stability related to the value of $\delta_{cr}$ and the reduction in standard deviations in performances. *Second*, VCEB learns a Gaussian distribution; a mean but also an input-dependant covariance $e_i^\sigma(x)$. This covariance fits the uncertainty of a given sample: in a similar context, Yu et al. (2019) has shown that large covariances were given for difficult samples. Sampling from this input-dependant covariance performs better than using an arbitrary fixed variance shared by all dimensions from all extracted features from all samples, from 77.29 to 77.51.

**Conclusion** DICE benefits from both components: learning redundancy along with VCEB improves results, at almost no extra cost. We think CR can definitely be applied with deterministic encoders as long as the inputs of the discriminator are sampled from overlapping distributions in the features space. Future work could study new methods to select the variance in sampling. As compression losses yield additional hyper-parameters and may underperform for some architectures/datasets, learning only the conditional redundancy (without compression) could increase the applicability of our contributions.

## G IMPACT OF THE SECOND TERM IN THE NEURAL ESTIMATION OF CONDITIONAL REDUNDANCY

### G.1 CONDITIONAL REDUNDANCY IN TWO COMPONENTS

The conditional redundancy can be estimated by the difference between two components:

$$\hat{\mathcal{I}}_{DV}^{CR} = \frac{1}{|\mathcal{B}_{\text{J}}|} \sum_{(z_1,z_2,y) \in \mathcal{B}_{\text{J}}} \underbrace{\log f(z_1, z_2, y)}_{\text{Diversity}} - \log \left( \frac{1}{|\mathcal{B}_p|} \sum_{(z_1,z_2',y) \in \mathcal{B}_p} \underbrace{f(z_1, z_2', y)}_{\text{Fake correlations}} \right), \quad (11)$$

with $f = \frac{w}{1-w}$. In this paper, we focused only on the left hand side (LHS) component from equation 11 which leads to $\hat{\mathcal{L}}_{DV}^{CR}$ in equation 4. We showed empirically that it improves ensemble diversity and overall performances. LHS forces features extracted from the same input to be unpredictable from each other; to simulate that they have been extracted from two different images.

Now we investigate the impact of the right hand side (RHS) component from equation 11. We conjecture that RHS forces features extracted from two different inputs from the same class to create fake correlations, to simulate that they have been extracted from the same image. Overall, the RHS would correlate members and decrease diversity in our ensemble.

### G.2 EXPERIMENTS

These intuitions are confirmed by experiments with a 4-branches ResNet-32 on CIFAR-100, which are illustrated in Figure 12. Training only with the RHS and removing the LHS (the opposite of what is done in DICE) reduces diversity compared to CEB. Moreover, keeping both the LHS and the RHS leads to slightly reduced diversity and ensemble accuracy compared to DICE. We obtained $77.40 \pm 0.19$ with LHS+RHS instead of $77.51 \pm 0.17$ with only the LHS. In conclusion, dropping the RHS performs better while reducing the training cost.

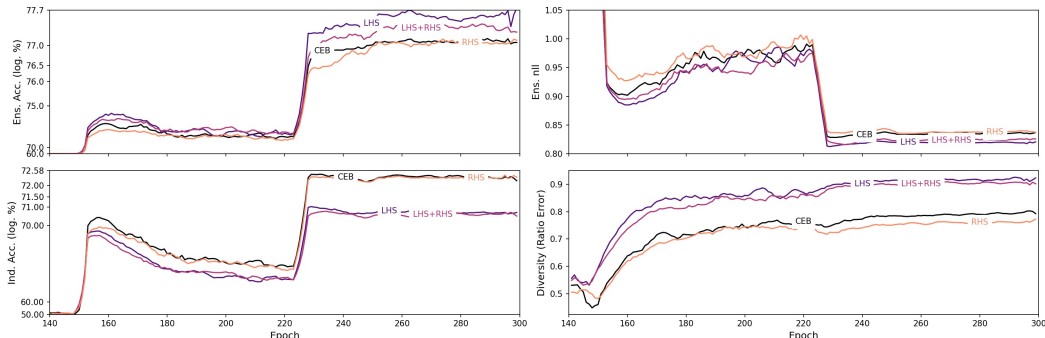

Figure 12: **Training dynamics and ablation study of components from equation 11**. Adding the RHS overall decreases ensemble performances, in terms of accuracy (upper left) or uncertainty estimation (upper right), when combined with CEB or DICE(=LHS). It decreases diversity (lower right) with no clear impact on individual accuracy (lower left).

## H  SOCIOLOGICAL ANALOGY

We showed that increasing diversity in features while encouraging the different learners to agree improves performance for neural networks: the optimal diversity-accuracy trade-off was obtained with a large diversity. To finish, we make a short analogy with the importance of diversity in our society. Decision-making in group is better than individual decision as long as the members do not belong to the same cluster. Homogenization of the decision makers increases vulnerability to failures, whereas diversity of backgrounds sparks new discoveries (Muldoon, 2016): ideas should be shared and debated among members reflecting the diversity of the society's various components. Academia especially needs this diversity to promote trust in research (Sierra-Mercado & Lázaro-Muñoz, 2018), to improve quality of the findings (Swartz et al., 2019), productivity of the teams (Vasilescu et al., 2015) and even schooling's impact (Bowman, 2013).

## I  LEARNING STRATEGY OVERVIEW

We provide in Figure 13 a zoomed version of our learning strategy.

## J  MAIN TABLE

Table 12 unifies our main results on CIFAR-100 from Table 1 and CIFAR-10 from Table 2.

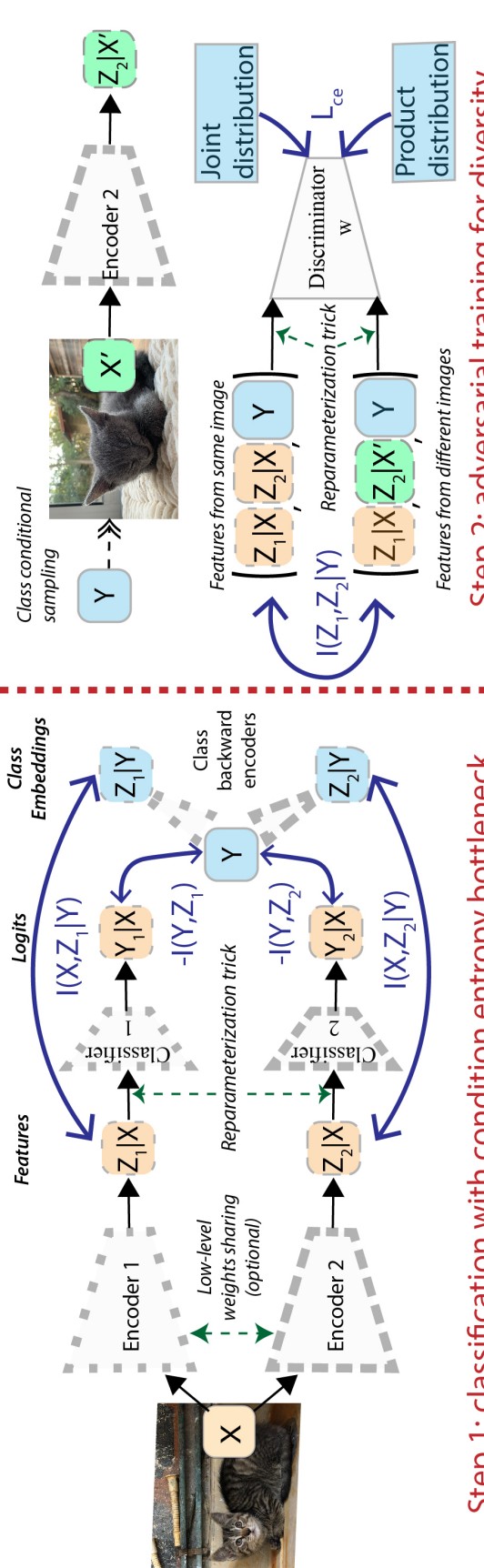

Figure 13: **Learning strategy overview**. Blue arrows represent training criteria: (1) classification with conditional entropy bottleneck applied separately on members 1 and 2, and (2) adversarial training to delete spurious correlations between members and increase diversity. $X$ and $X'$ belong to the same $Y$ for **conditional redundancy** minimization.

Table 12: **Ensemble classification accuracy (Top-1, %).**

| Method | | | CIFAR-100 | | | | | | | | | CIFAR-10 | |
|---|---|---|---|---|---|---|---|---|---|---|---|---|---|
| | Components | | ResNet-32 | | | | ResNet-110 | | WRN-28-2 | | | ResNet-32 | ResNet-110 |
| Name | Div. | I.B. | 3-branch | 4-branch | 5-branch | 4-net | 3-branch | 4-branch | 3-branch | 4-branch | 3-net | 4-branch | 3-branch |
| Ind. | | | 76.28±0.12 | | | 77.38±0.12 | 80.54±0.09 | 80.89±0.31 | 78.83±0.12 | 79.10±0.08 | 80.01±0.15 | 94.75±0.08 | 95.62±0.06 |
| ONE (Lan et al., 2018) | | | 75.17±0.35 | 75.13±0.25 | 75.25±0.22 | 76.25±0.32 | 78.97±0.24 | 79.86±0.25 | 78.38±0.45 | 78.47±0.32 | 77.53±0.36 | 94.41±0.05 | 95.25±0.08 |
| OKDDip (Chen et al., 2020b) | | | 75.37±0.32 | 76.85±0.25 | 76.95±0.18 | 77.27±0.31 | 79.07±0.27 | 80.46±0.35 | 79.01±0.19 | 79.32±0.17 | 80.02±0.14 | 94.86±0.08 | 95.21±0.09 |
| ADP (Pang et al., 2019) | Pred. | | 76.37±0.11 | 77.21±0.21 | 77.67±0.25 | 77.51±0.25 | 80.73±0.38 | 81.40±0.27 | 79.21±0.19 | 79.71±0.18 | 80.01±0.17 | 94.92±0.04 | 95.43±0.12 |
| IB (equation 8) | | VIB | 76.01±0.12 | 76.93±0.24 | 77.22±0.19 | 77.72±0.12 | 80.43±0.34 | 81.12±0.19 | 79.19±0.35 | 79.15±0.12 | 80.15±0.13 | 94.76±0.12 | 94.54±0.07 |
| CEB (equation 2) | | VCEB | 76.36±0.06 | 76.98±0.18 | 77.35±0.14 | 77.64±0.15 | 81.08±0.12 | 81.17±0.16 | 78.92±0.08 | 79.20±0.13 | 80.38±0.18 | 94.93±0.11 | 94.65±0.05 |
| IBR (equation 9) | R | VIB | 76.68±0.13 | 77.25±0.13 | 77.77±0.21 | 77.84±0.12 | 81.34±0.21 | 81.38±0.08 | 79.33±0.15 | 79.90±0.10 | 80.22±0.10 | 94.91±0.14 | 95.68±0.05 |
| CEBR (equation 10) | R | VCEB | 76.72±0.08 | 77.30±0.12 | 77.81±0.10 | 77.82±0.11 | 81.52±0.11 | 81.55±0.33 | 79.25±0.15 | 79.98±0.07 | 80.35±0.15 | 94.94±0.12 | 95.67±0.06 |
| DICE (equation 6) | CR | VCEB | **76.89**±0.09 | **77.51**±0.17 | **78.08**±0.18 | **77.92**±0.08 | **81.67**±0.14 | **81.93**±0.13 | **79.59**±0.13 | **80.05**±0.11 | **80.55**±0.12 | **95.01**±0.09 | **95.74**±0.08 |

