# OpenReview forum: "DICE: Diversity in Deep Ensembles via Conditional Redundancy Adversarial Estimation"
_ICLR.cc/2021/Conference — ICLR 2021 Poster_

### Official Review · AnonReviewer3 · 2020-10-29
**A novel application for IB objectives and information-theoretical quantities that improves diversity in ensembles**

**Rating:** 8
**Confidence:** 4

**Review:**

The paper introduces a new training criterion based on Information Bottleneck theory, which increases the diversity in an ensemble by minimizing the mutual information between latents of the different ensemble models. This leads to more diverse encodings that are useful for the task, which leads to better ensemble performance overall.

Overall, I’m scoring the paper with a weak accept. It provides a new principled application of Information Bottleneck objectives with good experimental results in a new area of application. I hope a baseline implementation of this new objective will be made available for others to use. However, I have a few concerns which I detail below which the authors will hopefully be able to resolve easily.

### Strengths

The paper is written very well. Overall, it is clear and engaging. The I-diagram helps understand the trade-offs within the objectives and makes it easy to see the information-theoretical equivalence of IBR and DICE (1).

The main conceptual idea is to add a term to minimize the (conditional) redundancy between latents of different ensemble members, which is expressed as the mutual information of the latent $I[Z_1;Z_2]$ in the case of an ensemble with two members. This term targets correlations between the latents which otherwise would add bias to the ensemble members.

The paper clearly describes the path from the general IB objective and the conceptual idea of reducing “redundancy” between (by minimizing to the CEB objective (Fischer, 2020) on to estimating the conditional redundancy.

It provides various experiments that show that this new method achieves better ensemble performance than similar approaches or independent ensembles that do not optimize for diversity.

### Concerns

1. The paper mentions that the second term on the RHS of (3) is not empirically necessary. However, they do not provide an ablation/experimental results that show the performance without dropping that term. For the sake of being principled, it would be nice to show evidence that dropping the term does not affect performance.

2. Using information-theoretic principles to increase diversity is great. This reviewer wonders if the IB objective is necessary at all however:
Conditional redundancy could also be minimized independently as the idea is not itself connected to the IB objective. $I[Z_1;Z_2|Y]$ could also be meaningfully computed for deterministic encoders, for example. As such, the applicability of the contribution could be increased by applying CR as regularizer to ensembles.

3. The authors write in 2.A.2: “The problem is that IBR ignores Y in the compression & redundancy terms which cause loss of necessary information” as motivation for the CEB. However, this is not true. Indeed, the CEB is equivalent to the regular IB objective for $\beta=2$. As such, there is not conceptual difference, above statement is misleading. The reviewer speculates that CEB might allow for stabler optimization compared to the regular IB objective because the terms in the regular IB objective are “fighting” against each other by both “including” $I[Y;Z]$ within its terms: one term to minimize it, the other to maximize it. As the two terms are usually estimated separately this can lead to instability and “infighting”. Another speculation is that VCEB is more flexible than DVIB in its variational approach.

---
Following the author's reply, I've raised my score from a 6 to a 8.

---

> ### Author Response · Authors · 2020-11-17
> **Response to Reviewer 3**
>
> We would like to thank the reviewer for these very interesting questions, that we will try to address below. We will soon open source the code in Pytorch and help for the reproducibility of our findings.
>
> Q1: “it would be nice to show evidence that dropping the [RHS] does not affect performance.”
>
> As a reminder, the neural estimation of the conditional redundancy leads to $\hat{\mathcal{I}}_{DV}^{CR}$, that is the difference between 2 components. The Left-Hand-Side (LHS) was underbraced with the term “diversity” while the Right-Hand-Side (RHS) was underbraced with the term “Fake correlations”.
> In this paper, we only use the LHS component and get rid of the RHS. This was backed up by our empirical findings, which are now included in newly added Figure 12 in Appendix G. We obtain $77.40$ with LHS+RHS instead of $77.51$ with only the LHS, with a 4-branches ResNet-32 on CIFAR-100. Training only with the RHS and removing the LHS (the opposite of what is done in DICE) reduces diversity compared to CEB. Moreover, keeping both the LHS and the RHS leads to slightly reduced diversity and ensemble accuracy compared to DICE.
>
> LHS improves ensemble diversity and overall performances. We argue that the LHS forces features extracted from the same input to be unpredictable from each other, to simulate that they have been extracted from two different images. On the contrary, the RHS correlates members and decreases diversity. As discussed in the paragraph “intuition” from subsection 2.B.2, we hypothesized that the RHS would force features extracted from two different inputs from the same class to create fake correlations; to simulate that they have been extracted from the same image.
> In conclusion, dropping the RHS performs better while reducing the training cost.
>
> Q2: “Using information-theoretic principles to increase diversity is great. Conditional redundancy could also be minimized [...] for deterministic encoders. [...] As such, the applicability of the contribution could be increased by applying CR as regularizer to ensembles.”
>
> The question is whether we could learn deterministic encoders with second order $I(Z_i; Z_j|Y)$ regularization (CR) without tackling first order $I(X; Z_i|Y)$ CEB components. This is a possible extension of our work that was mentioned briefly in our previous Appendix E.6. We ran additional experiments that are summarized below, and extensively detailed in the new Appendix F.2 and new Table 11.
>
> A first approach (feeding the discriminator $w$ with deterministic triples) increases diversity and reaches 77.09, compared to 76.78 for independent deterministic learning. A second approach, similar to instance noise [a], reached 77.33 by better protecting individual accuracies. In comparison, DICE obtains 77.51, showing that learning CR along with VCEB performs best, at almost no extra cost. In addition to the theoretical motivations, VCEB and CR work empirically in synergy.
> However, as compression losses may underperform for some architectures or datasets, learning only the conditional redundancy (without compression) could increase the applicability of our contributions in future works. This remark is now included in our conclusion.
>
> Finally, we would like to point out two possible explanations for the DICE success. First, the adversarial learning may be simplified as it only focuses on spurious correlations VCEB has not already deleted. Second, VCEB learns a distribution; a mean but also an input-dependant covariance $e_{i}^{\sigma}(x)$, that fits the uncertainty of a given sample $x$: [b] has shown that large covariances were given for difficult samples. Sampling in CR from this input-dependant covariance performs better than using an arbitrary fixed variance in Table 11.
>
> | Method   |      Top-1 Accuracy on CIFAR-100 on 4-branches ResNet-32|
> |----------|:-------------:|
> | Ind. |  76.78 |
> | Determ. + CR no sampling |  77.09   |
> | Determ + CR with sampling | 77.33 |
> |  DICE (VCEB + CR with input-dependant sampling) | 77.51 |
>
>
> [a] Sønderby, Casper Kaae, et al. "Amortised MAP Inference for Image Super-resolution." ICLR (2017).
>
> [b] Yu, Tianyuan, et al. "Robust person re-identification by modelling feature uncertainty." ICCV. 2019.
>
> Q3:  “CEB is equivalent to the regular IB objective [..] CEB might allow for stabler optimization compared to the regular IB objective”
>
> There is indeed an equivalence in [c] between the criteria for IB and CEB for appropriate $\beta$ and our sentence was misleading. The current submission has been refined and now simply states that: “The problem is that the compression and redundancy terms in IBR also reduce necessary information related to $Y$”.
> Our experiments are consistent with your intuitions. We found VCEB to lead to more stable training than DVIB: different variational approximations yield different training losses.
>
> [c] Fischer, Ian. "The conditional entropy bottleneck." arXiv (2020).

---

> > ### Comment · AnonReviewer3 · 2020-11-23
> > **Response to Rebuttal**
> >
> > Thank you very much for your detailed reply and the updates to the paper incl the appendix. I'm satisfied with the changes and will update my score accordingly.
> >
> > ---
> > PS: for I-diagrams, you might want to check out: R. W. Yeung, "A new outlook on Shannon's information measures," in IEEE Transactions on Information Theory, vol. 37, no. 3, pp. 466-474, May 1991, doi: 10.1109/18.79902.

---

> > > ### Author Response · Authors · 2020-11-24
> > > **Additional Citation**
> > >
> > > We thank you for your comment and this relevant reference (as it introduced I-Diagrams) that will be included in Section 2.A.

---

### Official Review · AnonReviewer1 · 2020-10-29
**Increasing ensemble performance by reducing conditional mutual information**

**Rating:** 6
**Confidence:** 3

**Review:**

## Summary
This paper introduces a training procedure for ensembles of neural networks that improves intra-member diversity to achieve better accuracy and calibration.

## Originality
This paper augments the VIB training objective of (Alemi et al., 2017) with two modifications:
- adding a mutual information penalty between encodings of the same input by different ensemble members
- conditioning the mutual information between encodings on the _input label_

The use of mutual information in training neural networks is not novel; indeed, as the authors point out, using mutual information between input and encoding; encoding and output during training is common. However, explicitly reducing mutual information between separate feature representations is, to the extend that I am aware of, a novel and intuitive contribution.

## Significance
This paper addresses a well-known, yet still only partially understood, fundamental problem in ensemble learning: improved diversity between ensemble members is correlated with improved ensemble performance, even though requiring diversity within logits reduces each member's accuracy on what should often be easy-to-predict inputs.

The authors address this paradox by choosing instead to favor diversity within the ensemble member's feature spaces, using a mutual-information based loss, and introduce approximations to efficiently minimize this loss.

## Clarity
I found the beginning of the paper very clear and well exposed; however, section 2 requires careful reading and several passes to understand completely; given that it contains the crucial definition of this paper's main contribution, this made the overall understanding of this paper more complicated.

I would recommend assigning more space to the equations, and perhaps removing Figure 3 (which, even on a screen for a non color-blind person, was difficult to parse) to allocate more space to defining crucial components of section 2.B.1.

I am very open to improving my score as these concerns are addressed.

## Pros/Cons

The idea behind this paper is both intuitive and elegant; however, I had a difficult time following anything more than the high-level explanation for VCEB. Although this may in part be my lack of familiarity with recent results, I found the explanation of this central part of the paper confusing.

- The appendix E.4 was helpful, but I would strongly recommend that the authors provide some clarifications in the main text: namely, defining the encoder, backward decoder, and the other terms that define the training loss.

- For the empirical neural estimation: how does the discriminator $w$ scale with the size of the ensemble? I would imagine that as $M$ increases (or as the complexity of the neural networks themselves increases), $w$ scales as well.

- Minor: I understand that it is difficult to fit all experimental results within the paper; however, Table 1 really needs to be larger.

- Pro: the authors included significant ablation studies for each component introduced in their training loss, and show meaningful (outside of standard deviation) improvement over several important baselines.

Finally, the authors may be interested in the recent work by A. Masegosa: [Learning under Model Misspecification: Applications to Variational and Ensemble methods](https://arxiv.org/pdf/1612.00410.pdf), which provides a theoretical justification for the use of diverse ensembles in improving predictive accuracy and calibration.

---

> ### Author Response · Authors · 2020-11-17
> **Response to Reviewer 1**
>
> We appreciate that you read our paper closely and highlight its possible theoretical and practical impacts. We will try to address your concerns below. To put it simply, we provided additional details and tried to improve the overall layout.
>
> Q1: “Allocate more space to defining crucial components of section 2.B.1”
>
> Section 2 is indeed the most challenging part of our argumentation. We build upon the work from Fischer [a] in which they explained in detail the variational approximation. These computations were summarized in Appendix E.4. In subsection 2.B.1, we quickly introduced the different components in the same order that they appeared in Equation 2, the VCEB loss. Previous explanations lacked a proper practical description and intuition of each element. That’s why we added the following paragraph in subsection 2.B.1 of the updated version:
>
> Practically, we parameterize all distributions with Gaussians. The encoder $e_i$ is a traditional neural network features extractor (e.g. ResNet-32) that learns distributions (means and covariances) rather than deterministic points in the features space. That's why $e_i$ transforms an image into 2 tensors; a features-mean $e_i^{\mu}(x)$ and a diagonal features-covariance $e_i^{\sigma}(x)$ each of size $d$ (e.g. $64$). The classifier $c_i$ is a dense layer that transforms a features-sample $z$ into logits to be aligned with the target $y$ through conditional cross entropy. $z$ is obtained via the reparameterization trick. Finally, the backward encoder $b_i$ is implemented as an embedding layer of size ($K$, $d$) that maps the $K$ classes to class-features-means $b_i^{\mu}(z|y)$ of size $d$, as we set the class-features-covariance to $1$. The Gaussian parametrization also enables the exact computation of the $KL$, that forces (1) features-mean $e_i^{\mu}(x)$ to converge to the class-features-mean $b_i^{\mu}(z|y)$ and (2) the predicted features-covariance $e_i^{\sigma}(x)$ to be close to $1$.
>
> We hope that these additional practical details clarify the argumentation.
>
> [a] Fischer, Ian. "The conditional entropy bottleneck." arXiv (2020).
>
>
> Q2: “For the empirical neural estimation: how does the discriminator scale with the size of the ensemble?”
>
> One key practical strength of our paper is the use of only one discriminator $w$ for the $\binom{M}{2}$ neural approximations of pairwise interactions, as stated in Section 2.C. The number of weights in $w$ scales linearly with the number of members $M$ and the embedding size $d$ as $w$’s input grows linearly. $w$’s hidden size and the number of hidden layers remain fixed. This methodology scaled for network-ensembles at least up to 10 ResNet-32 where d=64, and at least up to 5 ResNet-110 where d=256.
> In more details, $w$ is a multi-layer perceptron with 4 layers of size {256, 256, 100, #classes}. This architecture was described in Appendix B.4 and was inspired by Tables 3 and 4 from [b]. $w$ takes as input M tensors of size $d$ and maps them to the first layer of size 256: therefore its first matrix grows with $M$ * $d$ * $256$ .
> We verify that our discriminator with 256 hidden size remains calibrated (as shown on Figure 11) in all our setups. As stated in [b], calibration is important to obtain consistent estimation of conditional mutual information through neural estimation. We tried to widen the hidden size of $w$ to $M*d$ (rather than a fixed 256) but it has not consistently improved overall results. That’s why an additional member in the ensemble only adds $256$ * $d$ trainable weights in $w$. This important point was only briefly underlined in Section 2.C and is now better highlighted.
>
> [b] Mukherjee, Sudipto, Himanshu Asnani, and Sreeram Kannan. "CCMI: Classifier based conditional mutual information estimation." PMLR, 2020.
>
> Q3: Recommendations about general layout and Table 1 in particular
>
> Previous Table 1 is now divided into two separate tables: the main one dedicated to CIFAR-100 (new Table 1), the second one to CIFAR-10 (new Table 2). The unified Table 12 can still be found in Appendix J.
> Moreover, the general layout has been improved. We assigned more spaces to highlight the main equations in Section 2, which we hope are more quickly visible at first glimpse now. Moreover, we edited and widened Figure 3 for improved clarity, while including a zoomed version in Figure 13 in Appendix I.
>
>
> Citation: Thank you for mentioning Masegosa [c]. It is now included in our new introduction as they also tackle the trade-off between individual accuracy and ensemble diversity. Their theoretical second-order PAC-Bayes bound leads to an interesting diversity-inducing learning strategy that we included in our related work. They stated that “random initialization [...] is one of the key ingredients to make them diverse”: our work demonstrates that explicitly reducing conditional redundancy is another key.
>
> [c] Andres R. Masegosa. "Learning under Model Misspecification: Applications to Variational and Ensemble methods." Neurips, 2020.

---

### Official Review · AnonReviewer2 · 2020-10-29
**information theory for ensembles**

**Rating:** 7
**Confidence:** 4

**Review:**

Summary:
This paper proposes a method of learning ensembles that adhere to an "ensemble version" of the information bottleneck principle. Whereas the information bottleneck principle says the representation should avoid spurious correlations between the representation (Z) and the training data (X) that is not useful for predicting the labels (Y), i.e. I(X;Z) or I(X;Z|Y), this paper proposes that ensembles should additionally avoid spurious correlations between the ensemble members that aren't useful for predicting Y, i.e. I(Z_i; Z_j| Y). They show empirically that the coefficient on this term increases diversity at the expense of decreasing accuracy of individual members of the ensemble.

Evaluation:
Overall, this is a strong submission and should be accepted. It is well-written and thoroughly cites and explains prior work - in several cases, I find their explanations of prior work more clear than the original papers. It makes a clear and (as far as I know) novel contribution in terms of the approach, and the empirical results seem thorough and fair. In particular, the experiments for classification accuracy are thorough and make meaningful improvements over information bottleneck baselines (VIB and VCEB). I would (of course) like to see ImageNet experiments, but given that these are ensembles of relatively large networks (ResNet32 and ResNet110) and they include CIFAR-100 results on a number of architecture variants and a handful of baselines, I think these experiments are sufficient for acceptance. I don't have enough expertise in the uncertainty and calibration results to evaluate whether those empirical results are especially strong.

Suggestions for improvement:
- Page 4, Empirical Neural Estimation: I'm not clear on one thing here - when you are sampling B_p from the product distribution, you talk about p(z_2|y) as though you have access to the true distribution here, but I'd expect that here you would only have the backward encoder distribution b(z_2|y), a variational approximation to the true p(z_2|y). (And then when you're optimizing w towards the optimal w*, you would be optimizing w towards the optimal w* for that approximate distribution, not the true distribution?) I think this is what you are alluding to when you call this the "false product distribution" in the next paragraph, but it wasn't clear. I believe this is inconsequential because you then drop this term entirely, but this point could use better explanation.
- It would be useful to include an explanation of how you chose the value of beta for the VIB and VCEB baselines. (I see in a section titled "Scheduling" that you anneal beta according to a specific schedule, but I wasn't clear on why or whether you tuned this at all.)
- The ensemble experiment results (Section 4C and Table 2) are really hard to understand and interpret. The acronyms in the table should be spelled out (maybe in the caption), and in Section 4C when you mention ECE and TACE for the first time, you should write out the acronyms and also cite where these metrics were introduced.
- I'd suggest citing the ensemble + uncertainty results from Ovadia et al. 2019 (https://arxiv.org/pdf/1906.02530.pdf).

---

> ### Author Response · Authors · 2020-11-17
> **Response to Reviewer 2**
>
> We would like to thank the reviewer for these positive comments. We address your concerns in detail below.
>
>
> Q1: “Empirical neural estimation: [...] sampling $\mathcal{B}_{p}$ from the product distribution”
>
> For the training of the discriminator $w$ with $L_{ce}$ (in subsection 2.B.2), we sample from the product distribution $p(z_1, y)p(z_2|y)$. You proposed to approximate $p(z_2|y)$ by its variational approximation $b(z_2|y)$: we have tried and it worked, but not as consistently as what we have done.
> In fact, we can estimate the true distribution $p(z_2|y)$ with a simple trick: by taking the features $z_2$ extracted from another image $x’$ that has the same class $y$ as $x$. Indeed, $p(z_2|x’ \text{ belongs to } y)$ is an empirical estimate of $p(z_2|y)$ on the classification training dataset. In Figure 4, that’s the reason why the two images both belong to the same class $y$. The discriminator is therefore given as input the triple from the estimated product distribution: $(z_1|x, z_2|x’, y)$, where $x$ and $x’$ belongs to the class $y$.
> However, such an example $x’$ belonging to the same class as $x$ may not be found in the current batch. Indeed, the batch size (=128) is in the same order of magnitude as the number of classes (=100 for CIFAR-100). That’s why we keep a memory from previous batches, as explained in the ‘sampling’ paragraph of Appendix B.4. This features' memory is of size $M$ * $d$ * $K$ * $4$, where $M$ is the number of members, $d$ the features dimension and $K$ the number of classes. It is updated at the end of each step.
> We have used these terms (true/false) for the joint and product distributions, by analogy with the classical generative adversarial framework (true real image/fake generated image). This inappropriate use has been fixed in the new submission to avoid any misunderstanding.
>
>
> Q2: “The value of beta for the VIB and VCEB”
>
> As detailed in paragraph “Scheduling” from Appendix B.2, we employ the jump-start annealing method for scheduling our IB losses. It was first introduced in [a] and is now common practice for information bottleneck approaches such as recent works [b] and [c]. [c] stated that "It makes it much easier to train models at low [$\beta_{ceb}$] , and appears to not negatively impact final performance".
> As the other hyperparameters, the scheduling was adapted on a validation dataset made of 5% of the training dataset, for a 4-branches ResNet-32 ensemble trained on VCEB. For CIFAR-10, we therefore took the scheduling from [c], except that we widened the intervals to make the training loss $D_kl(e_m, b_m)$ decrease more smoothly. We trained several different models for different values of $\log(\beta_{ceb})$, the results were robust on the interval [1.7, 2.2] and we took the round number 2.0. No standard scheduling was available for CIFAR-100. As it is more difficult than CIFAR-10, we added additional jump-epochs with lower $\beta_{ceb}$ with wider intervals. $\beta_{ib}$ were deduced from the equivalence from [d]: $\beta_{ib} = \beta_{ceb} + 1$.
> These schedulings have been used in all our setups, without and with redundancy losses, for ResNet-32, ResNet-110 and WRN-28-10, for $M$ from 1 to 10 members.
>
> [a] Wu, Tailin, et al. "Learnability for the information bottleneck." PMLR, 2020.
>
> [b] Wu, Tailin, and Ian Fischer. "Phase Transitions for the Information Bottleneck in Representation Learning." ICLR 2019.
>
> [c] Fischer, Ian, and Alexander A. Alemi. "CEB Improves Model Robustness." arXiv (2020).
>
> [d] Fischer, Ian. "The conditional entropy bottleneck." arXiv (2020).
>
>
> Q3: “The ensemble experiment results (Section 4C and Table 2) are really hard to understand and interpret.”
>
> [e] established new standards for quantifying performances of in-domain calibration and uncertainty for image classification. [e] argues that comparisons of metrics such as negative log likelihood should be made after temperature scaling [f]. In section 4.C, we apply this procedure.
> Due to the 8-page constraint, details, full acronyms and appropriate citations of these metrics were in previous Appendix D.2. They now introduce Experiment 4.C. We also did a more detailed analysis of our results. Moreover, to simplify the readability of the associated Table, we only kept scores after temperature scaling. Those before temperature scaling were moved to Appendix A.6 in Table 9.
>
> [e] Ashukha, Arsenii, et al. "Pitfalls of In-Domain Uncertainty Estimation and Ensembling in Deep Learning." ICLR. 2019.
>
> [f] Guo, Chuan, et al. "On Calibration of Modern Neural Networks." ICML. 2017.
>
>
> Citation: Thank you for reminding us to cite the work from Ovadia [g], which has been a great motivation to study deep ensembles.
>
> [g] Ovadia, Yaniv, et al. "Can you trust your model's uncertainty? Evaluating predictive uncertainty under dataset shift." Neurips. 2019.

---

> > ### Comment · AnonReviewer2 · 2020-11-17
> > **gender pronouns**
> >
> > Your gender pronoun in this response is sexist (not all reviewers are male).

---

> > > ### Author Response · Authors · 2020-11-17
> > > **Apologies for the use of a gendered pronoun**
> > >
> > > We sincerely apologize for this mistake, which runs contrary to what we believe in and our call for diversity in research (in Appendix H). We understand and share your point of view. That was an unfortunate mistranslation. The gendered pronoun ("his") was edited in the previous response.

---

### Official Review · AnonReviewer4 · 2020-10-30
**Interesting idea with nice theoretical part, however limited experimental validation**

**Rating:** 6
**Confidence:** 3

**Review:**

This paper  addresses the problem of training an ensemble of learning algorithms so to boost the diversity among single learners while preserving the accuracy of each learner.
The paper is well written, the addressed problem is well exposed and relevant for this audience, the literature review is more than adequate.
The core idea and main contribution of this paper is to enforce diversity in the fature space rather than in the outputs.
The formalization of the approximated solution in Sec 2B has merit, however it is not clear if this is a first or higher order approximation and in the case why not considering an higher order ?
Results on small datasets show some improvements, in some cases above 1%; yet, it is not clear how the proposed technique would perform on larger datasets such as ImageNet, which downtones a bit the value of this contribution.

---

> ### Author Response · Authors · 2020-11-17
> **Response to Reviewer 4**
>
> Thank you for your review and positive comments about the problem setting and for raising this question about higher order approximations. We detail our answer below, and also in Appendix F.1.
>
> Our training criterion was derived from information bottleneck principles (in section 2.A) for an ensemble of $M=2$ members, and was transformed into a tractable loss (in section 2.B) using two distinct methods: a variational approximation VCEB for first order $I(Z_i; X|Y)$ (in subsection 2.B.1) and a neural estimator for second order $I(Z_i; Z_j|Y)$ (in subsection 2.B.2). Then, we generalize our training criterion for M> 2 models (in section 2.C). This is where DICE indeed approximates the information between extracted features by assuming only pairwise interactions while truncating higher orders. It is common practice, for example in the feature selection literature [a,b,c].
>
> More precisely, let’s define the order of an information interaction as the number of different extracted features involved. The first order is $I(Z_i; X|Y)$ and is tackled by the CEB component. The second order is $I(Z_i; Z_j|Y)$ and is tackled by the pairwise conditional redundancy component. The third order is $I(Z_i; Z_j, Z_k|Y)$. We could up to a M-th order.
>
> Higher order interactions may capture more complex correlations. The main reason to truncate them is computational, as the number of components would grow exponentially and add significant additional cost in training. Another reason is empirical, the additional hyperparameters may be hard to optimize. Tackling first and second order interactions already brought significant and consistent gain in many setups, for many architecture variants, in terms of accuracy and uncertainty estimation. The complete analysis of these higher order interactions has huge potential and could lead to a future research project.
>
> [a] Battiti, Roberto. "Using mutual information for selecting features in supervised neural net learning." IEEE Transactions on neural networks 5.4 (1994)
>
> [b] Fleuret, François. "Fast binary feature selection with conditional mutual information." Journal of Machine learning research 5.Nov (2004)
>
> [c] Brown, Gavin. "A new perspective for information theoretic feature selection." Artificial intelligence and statistics. 2009.

---

### Author Response · Authors · 2020-11-17
**General Response to All Reviewers**

We would like to sincerely thank you for your thoughtful remarks and overall positive comments. We much appreciate that you all understand our ideas and found them interesting. We have tried to answer your questions individually. Your feedback enabled us to update a new version of the paper.

1. Clarity. Overall, we modified the layout by leveraging the additional 9-th page. We assigned more space to the main equations in Section 2 and further detailed the components of VCEB in Section 2.B.1 (Reviewer1 Question1, R1Q1). We explicitly state in Section 2.C that our discriminator’s size grows linearly with $M$ and $d$ (R1Q2). We divided the Table 1 into two smaller Tables (R1Q3). We describe in more detail the Experiment 4.C whose Table was simplified (R2Q3). We refined some choices of words and included new relevant citations.

2. Additional appendices. In Appendix F.1, we explain that DICE is a second-order approximation in terms of information interactions (R4). In Appendix F.2 and Table 11, we apply our diversity regularization to deterministic encoders (R3Q2). Appendix G and Figure 12 empirically motivate the removal of the RHS from our neural estimation (R3Q1).

---

### Decision · Program_Chairs · 2021-01-07
**Final Decision**

**Decision:**

Accept (Poster)

**Comment:**

This paper proposes a new method of learning ensembles of neural networks based on the Information Bottleneck theory, which increases the diversity in an ensemble by minimizing the mutual information between latent features of the different ensemble models. It shows promising results on classification, calibration and uncertainty estimation. The paper is well-written and the comments were properly addressed.